# MoE++: Accelerating Mixture-of-Experts Methods with Zero-Computation Experts

**Peng Jin**[1,2]**, Bo Zhu**[3]**, Li Yuan**[1,2,4] ✉ **, Shuicheng Yan**[3,5] ✉

[1]Pengcheng Laboratory
[2]School of Electronic and Computer Engineering, Shenzhen Graduate School, Peking University
[3]Kunlun 2050 Research & Skywork AI
[4]Rabbitpre Intelligence    [5]National University of Singapore
jp21@stu.pku.edu.cn, yuanli-ece@pku.edu.cn
 **Code: https://github.com/SkyworkAI/MoE-plus-plus**

## Abstract

In this work, we aim to simultaneously enhance the effectiveness and efficiency of Mixture-of-Experts (MoE) methods. To achieve this, we propose MoE++, a general and heterogeneous MoE framework that integrates both Feed-Forward Network (FFN) and zero-computation experts. Specifically, we introduce three types of zero-computation experts: the zero expert, copy expert, and constant expert, which correspond to discard, skip, and replace operations, respectively. This design offers three key advantages: (i) **Low Computing Overhead**: Unlike the uniform mixing mechanism for all tokens within vanilla MoE, MoE++ allows each token to engage with a dynamic number of FFNs, be adjusted by constant vectors, or even skip the MoE layer entirely. (ii) **High Performance**: By enabling simple tokens to utilize fewer FFN experts, MoE++ allows more experts to focus on challenging tokens, thereby unlocking greater performance potential than vanilla MoE. (iii) **Deployment Friendly**: Given that zero-computation experts have negligible parameters, we can deploy all zero-computation experts on each GPU, eliminating the significant communication overhead and expert load imbalance associated with FFN experts distributed across different GPUs. Moreover, we leverage gating residuals, enabling each token to consider the pathway taken in the previous layer when selecting the appropriate experts. Extensive experimental results demonstrate that MoE++ achieves better performance while delivering $1.1\sim2.1\times$ expert forward throughput[†] compared to a vanilla MoE model of the same size, which lays a solid foundation for developing advanced and efficient MoE-related models.

## 1 Introduction

Large Language Models (LLMs) (Brown et al., 2020; OpenAI, 2022; Ouyang et al., 2022; Chowdhery et al., 2023; Achiam et al., 2023) have achieved substantial advancements, primarily attributed to the expansion of training data and a significant increase in model parameters. However, the pursuit of ever-larger model sizes incurs prohibitive computational costs. Therefore, the Mixture-of-Experts (MoE) architecture (Jacobs et al., 1991; Zhou et al., 2022; Roller et al., 2021), which allows for parameter scaling while keeping computational costs manageable, has become a preferred solution. The recent incorporation of MoE architectures into Transformers (Vaswani et al., 2017) has enabled the effective scaling of language models to impressive sizes, resulting in exceptional performance (Team, 2024; Dai et al., 2024; Jiang et al., 2024; Shen et al., 2024; Wei et al., 2024). These achievements underscore the significant potential and promise of MoE language models.

Most existing Mixture-of-Experts (MoE) methods (Du et al., 2022; Lewis et al., 2021; Rajbhandari et al., 2022; Zeng et al., 2024) typically activate a fixed number of Feed-Forward Networks (FFNs) for all tokens. In many works (Lepikhin et al., 2021; Xue et al., 2024), each token selects the top

---

[*]Corresponding author: Li Yuan, Shuicheng Yan.
[†]We define expert throughput as the throughput of FFN experts and zero-computation experts (if present).

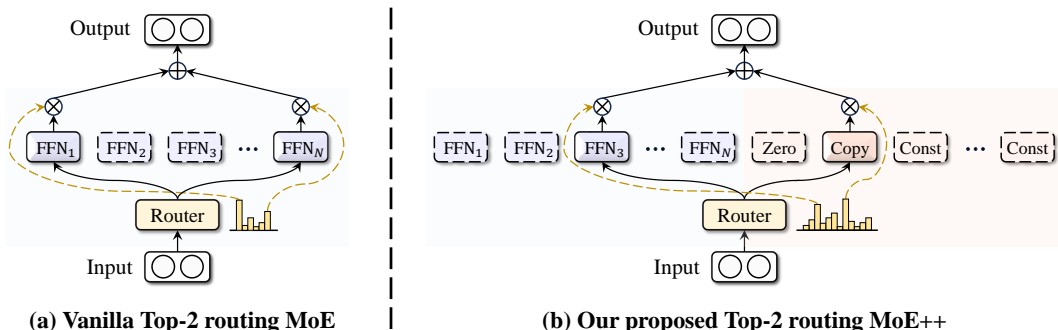

Figure 1: **A high-level comparison between the vanilla MoE and our proposed MoE++ architecture.** Subfigure (a) illustrates a standard MoE layer utilizing a Top-2 routing strategy, while subfigure (b) demonstrates the integration of zero-computation experts in MoE++. It is worth noting that these zero-computation experts require an almost negligible number of parameters, ensuring that the total parameter count for MoE++ is preserved at the same level as the vanilla MoE.

two FFNs and aggregates their outputs as the input for the subsequent layer. However, it is evident that not all tokens hold equal prediction difficulty in language tasks. For example, simple tokens, such as punctuation marks like commas, may only require a single expert. Conversely, tokens that poorly align with the existing experts might potentially benefit from bypassing the MoE layer entirely. Drawing from this insight, we contend that the rigidly fixed mixing mechanism used in previous work leads to training and inference inefficiencies, ultimately resulting in sub-optimal model performance.

In this work, we propose a general and heterogeneous MoE framework, called MoE++. To achieve a flexible computation allocation, we introduce three types of zero-computation experts: the zero expert, which discards input; the copy expert, which replicates input; and the constant expert[‡], which substitutes input with a trainable vector. As shown in Fig. 1, unlike vanilla MoE methods that restrict each token to a fixed number of FFN experts, MoE++ allows each token to engage with a variable number of FFN experts, receive adjustments through constant vectors, or even bypass the MoE layer entirely. This heterogeneous structure has a higher fitting ability by broadening the range of sub-network combinations with less computing overhead than vanilla MoE. Furthermore, we incorporate gating scores from the previous layer into the expert selection of the current layer. These gating residuals enable each token to consider its previous pathway when selecting the experts.

Starting with a modest scale of 0.6B parameters and expanding to 7B, extensive experimental results show that our MoE++ method significantly outperforms the vanilla MoE method by a substantial margin. It is worth noting that when scaled to 7B parameters and trained from scratch with 1T tokens, the MoE++ model achieves better performance than OpenMoE-8B/32E (Xue et al., 2024), a larger MoE model trained from scratch with 1.1T tokens. Meanwhile, the MoE++ model requires only about 57% of the computational cost of OpenMoE-8B/32E. More encouragingly, MoE++ allows simple tokens to utilize fewer FFN experts, freeing up more FFN experts to focus on challenging tokens. This results in both **Reduced Computation** and **Enhanced Performance**. Moreover, since the memory overhead of zero-computation experts is negligible, we can deploy all zero-computation experts on each GPU, eliminating significant communication overhead and expert load imbalance. Therefore, MoE++ is highly **Deployment-Friendly**. Extensive experiments show that MoE++ achieves approximately a 15%~111% increase in expert forward throughput compared to a vanilla MoE model of the same size. The main contributions of this work are summarized as follows:

- **Zero-computation experts.** To the best of our knowledge, we are the first to propose zero-computation experts for the MoE architecture. By introducing zero-computation experts, MoE++ has a higher fitting ability with less computing overhead than vanilla MoE.

- **Gating residuals.** We introduce gating residuals, which empower each token to consider its previous pathway when selecting the appropriate experts in the current MoE++ layer.

- **Flexible computation allocation.** MoE++ optimizes computation allocation by assigning fewer FFN experts to simple tokens, allowing more FFN experts to be dedicated to challenging tokens. Extensive experiments demonstrate that MoE++ not only enhances overall

---

[‡]Constant experts involve negligible computation, so we also consider them as zero-computation experts.

performance but also delivers up to $2\times$ expert forward throughput compared to vanilla MoE methods, laying a foundation for developing advanced and efficient language models.

## 2 RELATED WORK

**Large Language Models.** Large language models (Kenton & Toutanova, 2019; Radford et al., 2019; Raffel et al., 2020; Vaswani et al., 2017) have shown remarkable capabilities across a wide range of open-ended tasks and have extended their utility to include multimodal conversations (Liu et al., 2024c;b; Jin et al., 2024; Lin et al., 2023; 2024; Liu et al., 2024a), marking significant progress toward achieving general artificial intelligence. This success is largely attributed to the expansion of training data and the substantial increase in model parameters. Recently, various approaches (Team, 2024; Brown et al., 2020; Ouyang et al., 2022; OpenAI, 2022) have been proposed to scale model capacity and enhance performance, with efforts successfully expanding models to billions of parameters through different forms of model parallelism. However, the pursuit of ever-larger model sizes incurs prohibitive computational costs. Therefore, to enable the continued scaling of neural networks, improving the efficiency of model training and serving has become a critical research focus.

**Mixture-of-Experts Models.** The Mixture-of-Experts (MoE) method has been proposed to increase the capacity of a deep neural network without raising computational costs. The MoE method activates only a subset of parameters for each input, with these active parameters referred to as experts. Shazeer et al. (2017) introduces an MoE layer between LSTM layers, achieving impressive results in language modeling and machine translation benchmarks. Subsequently, the MoE layer is incorporated into the transformer architecture as a replacement for the feed-forward network layers. Switch Transformer (Fedus et al., 2022) simplifies the gating by selecting only the Top-1 expert per token, achieving better scaling compared to previous methods. Gshard (Lepikhin et al., 2021) improves the Top-2 expert routing strategy and significantly improves machine translation across 100 languages. Besides, BASE layer (Lewis et al., 2021), HASH layer (Roller et al., 2021), and Expert Choice (Zhou et al., 2022) explore ways to optimize MoE models for full utilization of their capacity.

## 3 METHODOLOGY

A standard Mixture-of-Experts (MoE) layer consists of $N$ expert networks $\boldsymbol{E} = \{E_1, E_2, ..., E_N\}$ and a router $G$ that activates the Top-K experts. Typically, the number of activated experts $K$ is fixed and much smaller than the total number of experts $N$. Formally, given the input token $\boldsymbol{x}$, the output token $\boldsymbol{y}$ of the MoE layer is the weighted sum of outputs from the $K$ activated experts:

$$\boldsymbol{y} = \sum_{i=1}^{N} g_i E_i(\boldsymbol{x}), \quad g_i = \begin{cases} \text{Softmax}\big(G(\boldsymbol{x})\big)_i, & \text{if } G(\boldsymbol{x})_i \in \text{Top-K}\big(\{G(\boldsymbol{x})_i | 1 \le i \le N\}\big). \\ 0, & \text{otherwise.} \end{cases} \quad (1)$$

**Vanilla MoE.** A vanilla MoE layer typically consists of multiple structurally identical experts, where each expert is a standard Feed-Forward Network (FFN). Besides, the router is usually implemented as a trainable weight matrix. Formally, the experts and router in a vanilla MoE layer can be defined as:

$$\boldsymbol{E} = \{\text{FFN}_1, \text{FFN}_2, ..., \text{FFN}_N\}, \quad G(\boldsymbol{x}) = \boldsymbol{W}\boldsymbol{x}, \quad (2)$$

where $\boldsymbol{W} \in \mathbb{R}^{N \times D}$ is the trainable weight matrix, and $D$ denotes the hidden size of the model. Since a fixed number of FFNs are activated in the vanilla MoE layer for both simple and challenging tokens, the vanilla MoE layer may be training and inference inefficient when processing simple tokens.

**MoE++ Overview.** Our proposed MoE++ is a general MoE framework that integrates both FFN and zero-computation experts. The core components of MoE++ are illustrated in Fig. 2.

### 3.1 ZERO-COMPUTATION EXPERTS

In MoE++, the redesigned expert architecture should satisfy specific criteria: (i) It should be as streamlined as possible to process simple tokens efficiently; (ii) To ensure a fair comparison with the vanilla MoE, the new expert should introduce an almost negligible number of parameters. Guided by these principles, we introduce zero-computation experts, each performing only the most fundamental operations. Specifically, we propose three types of zero-computation experts: the zero expert, copy expert, and constant expert, which correspond to discard, skip, and replace operations, respectively.

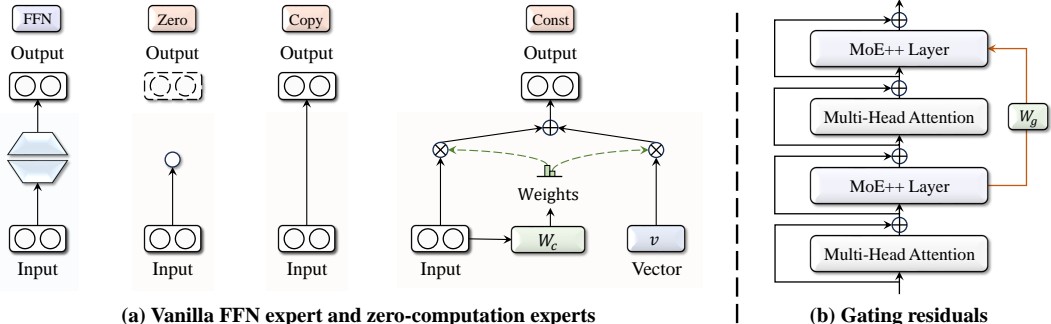

Figure 2: **The core components of the proposed MoE++.** Subfigure (a) illustrates the architectures of the FFN expert and zero-computation experts, while subfigure (b) shows the gating residuals which allow each token to consider its previous pathway to select the appropriate experts.

**Zero Experts.** The simplest zero-computation is to discard the current input. Given input token $\boldsymbol{x}$, the output of the zero expert is $\boldsymbol{0}$, which is formulated as:

$$E_{zero}(\boldsymbol{x}) = \boldsymbol{0}. \tag{3}$$

In essence, the presence of zero experts can degrade the Top-2 MoE++ layer to the Top-1 MoE++ layer. Specifically, if a zero expert is activated, its zero output makes the output of Top-2 MoE++ layers equivalent to that of the other expert alone. Therefore, in MoE++, the introduction of zero experts adds flexibility in handling both simple and challenging tokens simultaneously.

**Copy Experts.** Inspired by residual networks (He et al., 2016), we propose the copy expert, whose output is equal to the input and therefore equivalent to a shortcut:

$$E_{copy}(\boldsymbol{x}) = \boldsymbol{x}. \tag{4}$$

Intuitively, the copy expert offers the option to skip the current MoE++ layer. Specifically, if the input token is poorly aligned with the existing experts, it may benefit from bypassing the MoE++ layer.

**Constant Experts.** Neither zero experts nor copy experts contain trainable parameters, so their flexibility in handling input tokens is limited. To address this limitation, we propose constant experts, which replace the input token $\boldsymbol{x}$ with a trainable vector $\boldsymbol{v}$. However, a complete replacement would lose all input information. Therefore, we use a trainable weight matrix $\boldsymbol{W}_c$ to dynamically predict the proportion of the replacement. Formally, the output of copy experts can be defined as:

$$E_{const}(\boldsymbol{x}) = \alpha_1 \boldsymbol{x} + \alpha_2 \boldsymbol{v}, \quad [\alpha_1, \alpha_2] = \text{Softmax}(\boldsymbol{W}_c \boldsymbol{x}), \tag{5}$$

where $\boldsymbol{W}_c \in \mathbb{R}^{2 \times D}$ is the trainable weight matrix, and $D$ is the hidden size of the input token $\boldsymbol{x}$.

By assigning fewer FFN experts to simple tokens and dedicating more FFN experts to challenging tokens, MoE++ optimizes computation allocation. Therefore, MoE++ achieves better performance with less computation than vanilla MoE. Moreover, MoE++ significantly broadens the range of sub-networks. For instance, combining an FFN expert with a constant expert is equivalent to adjusting the output of the FFN expert using a trainable vector. Similarly, the combination of a zero expert with a copy expert allows the input token to bypass the current layer entirely.

## 3.2 PATHWAY-AWARE ROUTER

Since MoE++ contains heterogeneous experts, the design of the router becomes even more critical compared to vanilla MoE. To this end, we propose the pathway-aware router that considers the pathway taken in the previous layer when selecting the appropriate experts.

**Gating Residuals.** Similar to residual networks (He et al., 2016), we add the routing scores from the previous layer to the routing scores predicted by the current layer. Specifically, given the input token $\boldsymbol{x}^j$ of the $j_{th}$ layer with $N$ experts, we use a trainable transformation matrix $\boldsymbol{W}_g^j \in \mathbb{R}^{N \times N}$ to integrate the scores from the previous layer into the current layer:

$$G(\boldsymbol{x}^j) = \begin{cases} \boldsymbol{W}^j \boldsymbol{x}^j, & \text{if } j = 1, \\ \boldsymbol{W}^j \boldsymbol{x}^j + \boldsymbol{W}_g^j G(\boldsymbol{x}^{j-1}), & \text{if } j > 1, \end{cases} \tag{6}$$

Table 1: **Comparison of complexity between the proposed MoE++ and MoE.** Hyper-parameter $\tau$ controls the proportion of tokens allocated between zero-computation experts and FFN experts.

| Methods | # The Number of Tokens | # The Number of FFN Experts | # The Number of Zero-Computation Experts | Computation Complexity |
|---|---|---|---|---|
| MoE | $T$ | $N_{\text{FFN}}$ | $0$ | $\mathcal{O}(T)$ |
| **MoE++** | $T$ | $N_{\text{FFN}}$ | $N_{\text{ZC}}$ | $\mathcal{O}(\frac{\tau N_{\text{FFN}} T}{\tau N_{\text{FFN}} + N_{\text{ZC}}})$ |

where $\boldsymbol{W}^j \in \mathbb{R}^{N \times D}$ is the trainable weight matrix, and $D$ is the hidden size. These gating residuals effectively establish connections between MoE++ layers, therefore ensuring stable routing.

## 3.3 LOAD BALANCE DURING PRETRAINING

Training an MoE model directly often results in most tokens being dispatched to a small number of experts, leaving other experts insufficiently trained (Shazeer et al., 2017). Following previous works (Lepikhin et al., 2021; Xue et al., 2024; Dai et al., 2024; Wei et al., 2024), we apply the load balance loss and expert capacity to ensure a balanced load during pretraining.

**Heterogeneous Load Balance Loss.** In vanilla MoE methods, each expert is a standard Feed-Forward Network (FFN), so all experts are assigned the same number of tokens. However, in our proposed MoE++, the architecture and number of parameters in zero-computation experts and FFN experts differ significantly, making it sub-optimal to allocate the same number of tokens to both types of experts. To this end, we introduce a hyper-parameter $\tau$ to control the proportion of tokens allocated between zero-computation experts and FFN experts. Specifically, given the $t_{th}$ input token $\boldsymbol{x}_t$, the heterogeneous load balance loss $\mathcal{L}_b$ is formulated as:

$$\mathcal{L}_b = \sum_{i=1}^{N} \eta_i f_i P_i, \quad \eta_i = \begin{cases} 1, & \text{if Expert } i \text{ is an FFN expert,} \\ \tau, & \text{if Expert } i \text{ is a zero-computation expert,} \end{cases}$$

$$f_i = \frac{1}{T} \sum_{t=1}^{T} \mathbb{1}\left(\text{Token } \boldsymbol{x}_t \text{ selects Expert } i\right), \quad P_i = \frac{1}{T} \sum_{t=1}^{T} \text{Softmax}\left(G(\boldsymbol{x}_t)\right)_i,$$

(7)

where $T$ denotes the number of tokens. $N$ is the number of experts. $\mathbb{1}(*)$ denotes the indicator function. A smaller hyper-parameter $\tau$ means that more tokens are assigned to the zero-computation experts. In comparison, a larger $\tau$ means fewer tokens are allocated to the zero-computation experts.

**Expert Capacity.** Expert capacity is proposed to mitigate severe load imbalance by limiting the maximum number of tokens routed to each expert. Since MoE++ assigns different numbers of tokens to different types of experts, we also design varying expert capacities for each type of expert. For an MoE++ model with $N_{\text{FFN}}$ FFN experts and $N_{\text{ZC}}$ zero-computation experts, the total number of experts is $N = N_{\text{FFN}} + N_{\text{ZC}}$. Given the hyper-parameter $\tau$, the expert capacity is defined as:

$$C_i = \begin{cases} \gamma \frac{\tau T}{\tau N_{\text{FFN}} + N_{\text{ZC}}}, & \text{if Expert } i \text{ is an FFN expert,} \\ \gamma \frac{T}{\tau N_{\text{FFN}} + N_{\text{ZC}}}, & \text{if Expert } i \text{ is a zero-computation expert,} \end{cases}$$

(8)

where $\gamma$ is the preset capacity factor. $T$ is the number of tokens. Similarly, a smaller hyper-parameter $\tau$ means more capacity is allocated to the zero-computation expert. For both types of experts, if an expert is underutilized, its unused capacity is filled with padding tokens. Once an expert reaches capacity, any additional tokens assigned to that expert are dropped out, which means the additional tokens are passed directly to the subsequent Transformer (Vaswani et al., 2017) block.

**Total Training Objective.** Finally, the total training loss is the weighted sum of the cross-entropy loss $\mathcal{L}_{ce}$ and the heterogeneous load balance loss $\mathcal{L}_b$:

$$\mathcal{L} = \mathcal{L}_{ce} + \beta \mathcal{L}_b,$$

(9)

where $\beta$ is the trade-off hyper-parameter to mitigate the risk of routing collapse.

## 3.4 ANALYSIS OF EFFICIENCY

It is worth noting that zero-computation experts require a negligible amount of computing and communication costs to process a token. As shown in Tab. 1, for an MoE++ model with $N_{\text{FFN}}$ FFN

Table 2: **Sizes and architectures of MoE++ and vanilla MoE models.** "0.2B/0.6B" represents an architecture of an approximately 0.6B parameter, with 0.2B activated per token during inference. "1/1/2" denotes an MoE++ model with one zero expert, one copy expert, and two constant experts. In all models, two experts are activated at each layer.

| Methods | # Activated Params | # Layers | # Hidden Size | # Intermediate Size | # Heads | # Head Dim | # The Number of FFN Experts | # Zero/Copy/Constant Experts |
|---|---|---|---|---|---|---|---|---|
| MoE 0.6B/8E | 0.2B/0.6B | 12 | 768 | 2048 | 12 | 64 | 8 | - |
| MoE++ 0.6B/(8+4)E | ≤0.2B/0.6B | | | | | | | 1/1/2 |
| MoE 1B/16E | 0.2B/1B | 12 | 768 | 2048 | 12 | 64 | 16 | - |
| MoE++ 1B/(16+4)E | ≤0.2B/1B | | | | | | | 1/1/2 |
| MoE 2B/32E | 0.2B/2B | 12 | 768 | 2048 | 12 | 64 | 32 | - |
| MoE++ 2B/(32+8)E | ≤0.2B/2B | | | | | | | 1/1/6 |
| MoE 7B/16E | 1.2B/7B | 24 | 1536 | 4096 | 16 | 96 | 16 | - |
| MoE++ 7B/(16+4)E | ≤1.2B/7B | | | | | | | 1/1/2 |

experts, $N_{\text{ZC}}$ zero-computation experts and hyper-parameter $\tau$, its computational complexity is only $\frac{\tau N_{\text{FFN}}}{\tau N_{\text{FFN}} + N_{\text{ZC}}}$ that of MoE models with the same number of parameters.

# 4 EXPERIMENTS

## 4.1 EXPERIMENTAL SETUP

**Model Settings.** We use Megatron (Shoeybi et al., 2019), an open-source training code, as the training framework. We conduct training on a cluster with 4 nodes and 32 A100 GPUs. Tab. 2 summarizes the hyper-parameter settings of various MoE++ models. For example, "MoE++ 0.6B/(8+4)E" represents the architecture of an approximately 0.6B parameter MoE++ model with 8 FFN experts and 4 zero-computation experts. For a fair comparison, we also include the corresponding MoE model configurations with similar numbers of activated parameters per token during inference.

**Training Data and Tokenization.** MoE++ is trained exclusively on public datasets, making it accessible for academic research settings. Specifically, we sample from the **RedPajama** (Computer, 2023a), **Dolma** (Soldaini et al., 2024), and **Pile** (Gao et al., 2020) datasets according to different sampling probabilities. Please refer to Tab. D and Appendix B.1 for detailed sample ratios. We use the tokenizer of LLaMA2, which contains 65,536 vocabulary tokens.

**Training Hyper-Parameters.** The hyper-parameters for MoE++ are selected based on the common practice for dense language models. We replace all FFN layers in the transformer with MoE++ layers and set the Top-K to 2 for every layer, resulting in approximately twice the computation compared to a dense model. Please refer to Tab. E and Appendix B.2 for detailed training hyper-parameters.

**Evaluation Benchmarks.** We conduct comparative comparisons of MoE++ against vanilla MoE and dense models. Specifically, we use the lm-evaluation-harness package (Gao et al., 2024) to assess performance on an extensive suite of downstream tasks: (i) Following Pythia (Biderman et al., 2023) and Sheared-LLaMA (Xia et al., 2024), we report 0-shot accuracy on **ARC Easy (ARC-E)** (Clark et al., 2018), **LAMBADA** (Paperno et al., 2016), **LogiQA** (Liu et al., 2020), **PIQA** (Bisk et al., 2020), **SciQ** (Welbl et al., 2017), and **WinoGrande** (Sakaguchi et al., 2021). (ii) We also report the accuracy of tasks from the Open LLM Leaderboard (Beeching et al., 2023), including 10-shot **HellaSwag** (Zellers et al., 2019), 25-shot **ARC Challenge (ARC-C)** (Clark et al., 2018), and 5-shot **MMLU** (Hendrycks et al., 2021). (iii) Moreover, we report the exact match score for 32-shot **Natural Questions (NQ)** (Kwiatkowski et al., 2019) and the accuracy for 32-shot **BoolQ** (Clark et al., 2019).

## 4.2 MAIN RESULTS

**Comparisons to Vanilla MoE.** To conduct comparative evaluations of our proposed MoE++ against vanilla MoE models, we start with a modest scale of 0.6B parameters and expand up to 7B. Since the activated parameters are only about 0.2B for the smallest model, we select 9 simple benchmarks as the metric. As shown in Tab. 3, our proposed MoE++ consistently outperforms vanilla MoE models. Notably, MoE++ achieves a 15%~111% increase in expert forward throughput compared to a vanilla MoE model of the same size, while at the same time having better performance. The proposed MoE++ lays a solid foundation for developing advanced and efficient language models.

Table 3: **Comparisons between MoE++ and vanilla MoE models.** The training budget for all MoE++ and vanilla MoE models listed in the table below is 100B tokens.

| Methods | # $\tau$ | # Expert Forward Time (ms) | # Throughput Increase | Commonsense & Reading Comprehension | | | | |
|---|---|---|---|---|---|---|---|---|
| | | | | SciQ | PIQA | WinoGrande | ARC-E | HellaSwag (10) |
| MoE 0.6B/8E | - | 535.3 | - | **76.6** | 67.3 | 50.2 | 47.6 | 40.3 |
| MoE++ 0.6B/(8+4)E | 0.10 | 202.4 | 164.5% | 73.1 | 65.1 | 51.9 | 46.0 | 36.1 |
| MoE++ 0.6B/(8+4)E | 0.25 | 277.8 | 92.7% | 74.8 | 66.5 | 51.1 | 48.4 | 39.7 |
| MoE++ 0.6B/(8+4)E | 0.50 | 387.3 | 38.2% | 74.9 | **68.4** | 49.3 | 48.9 | 40.7 |
| MoE++ 0.6B/(8+4)E | 0.75 | 427.6 | 25.2% | 76.5 | 67.6 | 51.9 | **50.1** | **41.8** |
| MoE++ 0.6B/(8+4)E | 1.00 | 449.6 | 19.1% | 75.9 | 67.6 | **52.2** | 49.0 | 41.7 |
| MoE 1B/16E | - | 610.9 | - | **79.3** | 68.4 | **54.2** | 48.9 | 43.7 |
| MoE++ 1B/(16+4)E | 0.10 | 289.3 | 111.2% | 76.3 | 67.6 | 51.1 | 48.5 | 40.0 |
| MoE++ 1B/(16+4)E | 0.25 | 384.9 | 58.7% | 77.4 | 68.3 | 50.2 | 48.2 | 42.5 |
| MoE++ 1B/(16+4)E | 0.50 | 469.7 | 30.1% | 77.5 | 67.6 | 52.5 | 49.7 | 44.3 |
| MoE++ 1B/(16+4)E | 0.75 | 500.3 | 22.1% | 78.3 | **70.3** | 51.7 | 49.7 | **44.6** |
| MoE++ 1B/(16+4)E | 1.00 | 530.2 | 15.2% | **79.3** | 69.5 | 52.5 | **51.5** | **44.6** |
| MoE 2B/32E | - | 683.4 | - | 77.9 | 70.0 | 51.6 | 51.6 | 46.0 |
| MoE++ 2B/(32+8)E | 0.10 | 417.9 | 63.5% | 76.0 | 68.2 | 52.3 | 49.7 | 42.7 |
| MoE++ 2B/(32+8)E | 0.25 | 473.7 | 44.3% | 76.2 | 69.4 | 52.6 | 51.7 | 46.0 |
| MoE++ 2B/(32+8)E | 0.50 | 532.7 | 28.3% | 78.4 | 70.0 | **54.2** | 50.6 | 47.0 |
| MoE++ 2B/(32+8)E | 0.75 | 561.0 | 21.8% | 79.6 | **70.2** | 51.9 | 51.4 | 47.3 |
| MoE++ 2B/(32+8)E | 1.00 | 590.5 | 15.7% | **81.7** | 69.8 | 53.6 | **52.4** | **47.7** |
| MoE 7B/16E | - | 1859 | - | 78.3 | 72.6 | **58.8** | 53.1 | 61.3 |
| MoE++ 7B/(16+4)E | 0.75 | 1455 | 27.8% | **80.1** | **73.6** | 58.2 | **53.6** | **61.8** |

| Methods | # $\tau$ | # Expert Forward Time (ms) | # Throughput Increase | Continued | | LM | World Knowledge | Average |
|---|---|---|---|---|---|---|---|---|
| | | | | LogiQA | BoolQ (32) | LAMBADA | NQ (32) | |
| MoE 0.6B/8E | - | 535.3 | - | 25.3 | 50.9 | 38.7 | **1.5** | 44.3 |
| MoE++ 0.6B/(8+4)E | 0.10 | 202.4 | 164.5% | 27.5 | **57.6** | 33.2 | 0.3 | 43.4 |
| MoE++ 0.6B/(8+4)E | 0.25 | 277.8 | 92.7% | 27.8 | 55.9 | 38.3 | 1.2 | 44.9 |
| MoE++ 0.6B/(8+4)E | 0.50 | 387.3 | 38.2% | 27.3 | 56.5 | 37.4 | 1.1 | 44.9 |
| MoE++ 0.6B/(8+4)E | 0.75 | 427.6 | 25.2% | **28.7** | 54.2 | 38.7 | 1.1 | **45.6** |
| MoE++ 0.6B/(8+4)E | 1.00 | 449.6 | 19.1% | 26.9 | 46.1 | **39.6** | 1.1 | 44.5 |
| MoE 1B/16E | - | 610.9 | - | 27.5 | 41.2 | 42.5 | 2.1 | 45.3 |
| MoE++ 1B/(16+4)E | 0.10 | 289.3 | 111.2% | 26.4 | **60.8** | 40.5 | 0.9 | 45.8 |
| MoE++ 1B/(16+4)E | 0.25 | 384.9 | 58.7% | 27.2 | 57.2 | 40.2 | 2.1 | 45.9 |
| MoE++ 1B/(16+4)E | 0.50 | 469.7 | 30.1% | 27.0 | 50.7 | 42.5 | **2.4** | 46.0 |
| MoE++ 1B/(16+4)E | 0.75 | 500.3 | 22.1% | 27.6 | 46.3 | 43.9 | **2.4** | 46.1 |
| MoE++ 1B/(16+4)E | 1.00 | 530.2 | 15.2% | **27.8** | 52.2 | **45.3** | 1.8 | **47.2** |
| MoE 2B/32E | - | 683.4 | - | 28.1 | 41.2 | 43.9 | 2.9 | 45.9 |
| MoE++ 2B/(32+8)E | 0.10 | 417.9 | 63.5% | **28.7** | 58.0 | 39.1 | 1.8 | 46.3 |
| MoE++ 2B/(32+8)E | 0.25 | 473.7 | 44.3% | 28.6 | 54.3 | 42.3 | 1.8 | 47.0 |
| MoE++ 2B/(32+8)E | 0.50 | 532.7 | 28.3% | 26.7 | 45.1 | 43.5 | 2.2 | 46.4 |
| MoE++ 2B/(32+8)E | 0.75 | 561.0 | 21.8% | 27.7 | 48.2 | **46.0** | **3.6** | 47.3 |
| MoE++ 2B/(32+8)E | 1.00 | 590.5 | 15.7% | 25.7 | **58.4** | 44.9 | 3.0 | **48.6** |
| MoE 7B/16E | - | 1859 | - | 27.8 | 53.7 | **57.2** | 8.2 | 52.3 |
| MoE++ 7B/(16+4)E | 0.75 | 1455 | 27.8% | **28.3** | **56.0** | 57.0 | **9.0** | **53.1** |

**Comparisons to LLMs of Equivalent Activated Parameters.** Existing models usually employ substantial training budgets, such as OpenMoE-8B/32E with 1.1T tokens and TinyLlama-1.1B with 3T tokens. Similarly, as shown in Tab. 4, we extend the training budget of our MoE++ model to 1T tokens, aligning it with other models. We find that the MoE++ model delivers performance comparable to dense models that have 2 to 3 times the number of activation parameters. Notably, MoE++ outperforms OpenMoE-8B/32E, a larger MoE model trained from scratch with more tokens, while utilizing only approximately 57% of the computational cost of OpenMoE-8B/32. These results show that the proposed MoE++ method is a promising solution for training LLMs.

### 4.3 ABLATIVE ANALYSIS

**Effect of the hyper-parameter $\tau$ in Eq. 7 and Eq. 8.** The hyper-parameter $\tau$ controls the proportion of tokens distributed between zero-computation experts and FFN experts. To investigate the impact of the hyper-parameter $\tau$, we provide comparative evaluations in Tab. 3. As shown in Tab. 3, a smaller $\tau$ means that more tokens are assigned to the zero-computation experts with negligible computing costs, resulting in higher throughput. Conversely, a larger $\tau$ means fewer tokens are allocated to the zero-computation experts and generally have better performance. To balance computing costs and performance, we set the hyper-parameter $\tau$ to 0.75 by default.

Table 4: **Comparisons to LLMs of equivalent activated parameters.** "§" denotes that the model is not trained from scratch but is pruned and continue-tuned using the weights of LLaMA2-7B. Therefore, we gray out Sheared-LLaMA and LLaMA-MoE for a fair comparison. The results of OpenMoE-8B/32E are from its paper, so only partial task results are available.

| Methods | # Activated | Commonsense & Reading Comprehension | | | | | |
|---|---|---|---|---|---|---|---|
| | Params | SciQ | PIQA | WinoGrande | ARC-E | ARC-C (25) | HellaSwag (10) |
| LLaMA2-7B (Touvron et al., 2023) | 7B/7B | 93.7 | 78.1 | 69.3 | 76.4 | 53.0 | 78.6 |
| OPT-1.3B (Zhang et al., 2022) | 1.3B/1.3B | 84.3 | 71.7 | 59.6 | 57.0 | 29.7 | 54.5 |
| Pythia-1.4B (Biderman et al., 2023) | 1.4B/1.4B | 86.4 | 70.9 | 57.4 | 60.7 | 31.2 | 53.0 |
| TinyLlama-1.1B (Zhang et al., 2024) | 1.1B/1.1B | 88.9 | 73.3 | 58.8 | 55.3 | 30.1 | 60.3 |
| Sheared-LLaMA-1.3B§ (Xia et al., 2024) | 1.3B/1.3B | 87.3 | 73.4 | 57.9 | 61.5 | 33.5 | 60.7 |
| OPT-2.7B (Zhang et al., 2022) | 2.7B/2.7B | 85.8 | 73.7 | 60.8 | 60.8 | 34.0 | 61.5 |
| Pythia-2.8B (Biderman et al., 2023) | 2.8B/2.8B | 88.3 | 74.0 | 59.7 | 64.4 | 36.4 | 60.8 |
| INCITE-Base-3B (Computer, 2023b) | 3B/3B | 90.7 | 74.6 | 63.5 | 67.7 | 40.2 | 64.8 |
| Open-LLaMA-3B-v1 (Geng & Liu, 2023) | 3B/3B | 91.3 | 73.7 | 61.5 | 67.6 | 39.6 | 62.6 |
| Open-LLaMA-3B-v2 (Geng & Liu, 2023) | 3B/3B | 91.8 | 76.2 | 63.5 | 66.5 | 39.0 | 67.6 |
| Sheared-LLaMA-2.7B§ (Xia et al., 2024) | 2.7B/2.7B | 90.8 | 75.8 | 64.2 | 67.0 | 41.2 | 70.8 |
| OpenMoE-8B/32E (Xue et al., 2024) | 2.1B/8B | - | 74.2 | 60.3 | 64.1 | 30.3 | 45.5 |
| LLaMA-MoE-3.0B§ (Zhu et al., 2024) | 3.0B/7B | 89.9 | 77.5 | 63.6 | 66.8 | 40.9 | 70.8 |
| MoE++ 7B/(16+4)E | ≤1.2B/7B | 89.7 | **77.6** | 63.1 | 66.5 | **42.3** | **72.3** |

| Methods | # Active | Continued | | LM | World Knowledge | | Average |
|---|---|---|---|---|---|---|---|
| | Params | LogiQA | BoolQ (32) | LAMBADA | NQ (32) | MMLU (5) | |
| LLaMA2-7B (Touvron et al., 2023) | 7B/7B | 30.7 | 82.1 | 73.9 | 28.8 | 46.6 | 64.7 |
| OPT-1.3B (Zhang et al., 2022) | 1.3B/1.3B | 26.9 | 57.5 | 58.0 | 6.9 | 24.7 | 48.3 |
| Pythia-1.4B (Biderman et al., 2023) | 1.4B/1.4B | 27.3 | 57.4 | 61.6 | 6.2 | 25.7 | 48.9 |
| TinyLlama-1.1B (Zhang et al., 2024) | 1.1B/1.1B | 26.3 | 60.9 | 58.8 | 12.1 | 25.5 | 50.0 |
| Sheared-LLaMA-1.3B§ (Xia et al., 2024) | 1.3B/1.3B | 26.9 | 64.0 | 61.0 | 9.6 | 25.7 | 51.0 |
| OPT-2.7B (Zhang et al., 2022) | 2.7B/2.7B | 26.0 | 63.4 | 63.6 | 10.1 | 25.9 | 51.4 |
| Pythia-2.8B (Biderman et al., 2023) | 2.8B/2.8B | 28.0 | 66.0 | 64.7 | 9.0 | 26.9 | 52.6 |
| INCITE-Base-3B (Computer, 2023b) | 3B/3B | 27.7 | 65.9 | 65.3 | 14.9 | **27.0** | 54.8 |
| Open-LLaMA-3B-v1 (Geng & Liu, 2023) | 3B/3B | **28.4** | **70.0** | 65.4 | **18.6** | **27.0** | 55.1 |
| Open-LLaMA-3B-v2 (Geng & Liu, 2023) | 3B/3B | 28.1 | 69.6 | 66.5 | 17.1 | 26.9 | 55.7 |
| Sheared-LLaMA-2.7B§ (Xia et al., 2024) | 2.7B/2.7B | 28.9 | 73.7 | 68.4 | 16.5 | 26.4 | 56.7 |
| OpenMoE-8B/32E (Xue et al., 2024) | 2.1B/8B | - | 61.2 | - | - | - | - |
| LLaMA-MoE-3.0B§ (Zhu et al., 2024) | 3.0B/7B | 30.6 | 71.9 | 66.6 | 17.0 | 26.8 | 56.6 |
| MoE++ 7B/(16+4)E | ≤1.2B/7B | **28.4** | 68.3 | **67.4** | 18.1 | 24.6 | **56.2** |

(a) Results of MoE++ 0.6B/(8+{2+ $n_{const}$})E models    (b) Results of MoE++ 1B/(16+{2+ $n_{const}$})E models

Figure 3: **Ablation study on the number of constant experts.** We gradually increase the number of constant experts $n_{const}$ until the number of zero-computation experts is the same as that of FFN experts. All models are trained with a budget of 100B tokens, with the hyper-parameter $\tau$ set to 0.75.

**Effect of Each Zero-Computation Expert.**    In Tab. 5, we provide the ablation study on each zero-computation expert in "MoE++ 1B/(16+4)E" model. We find that constant experts improve the model more than zero experts and copy experts. We consider that it is due to the increased flexibility that constant experts provide in handling tokens. Specifically, zero experts output only empty features, copy experts replicate the input as output, and constant experts introduce additional trainable vectors to adjust the output. Our full model, which incorporates all three types of zero-computation experts, achieves the best performance, demonstrating their benefit for language models.

**Effect of the Gating Residuals.**    The gating residuals enable each token to consider the pathway taken in the previous layer when selecting the appropriate experts. To explore the influence of the gating residuals, we provide the ablation results in Tab. 6. We find that these simple gating residuals effectively establish connections between different MoE++ layers, thereby ensuring stable routing.

Table 5: **Ablation study on the impact of each zero-computation expert in "MoE++ 1B/(16+4)E" model.** All models are trained with a budget of 100B tokens, with the hyper-parameter $\tau$ set to 0.75.

| Zero Expert | Copy Expert | Constant Expert | Language Tasks | | | | | | | Average |
|---|---|---|---|---|---|---|---|---|---|---|
| | | | PIQA | WinoGrande | ARC-E | HellaSwag (10) | LogiQA | BoolQ (32) | LAMBADA | |
| | | | 68.4 | **54.2** | 48.9 | 43.7 | 27.5 | 41.2 | 42.5 | 46.6 |
| ✓ | | | 68.4 | 52.2 | 48.7 | 44.0 | 28.3 | 45.7 | 42.0 | 47.0 |
| | ✓ | | 69.4 | 52.1 | 50.0 | 44.0 | 27.7 | 45.1 | 42.3 | 47.2 |
| | | ✓ | 68.6 | 51.3 | 49.6 | 44.4 | **28.9** | 46.4 | 42.4 | 47.4 |
| ✓ | ✓ | | 68.6 | 52.6 | 49.2 | 44.1 | 28.1 | 45.8 | 41.8 | 47.2 |
| ✓ | | ✓ | 67.9 | 52.4 | **50.8** | 44.0 | 26.6 | 47.1 | 42.2 | 47.3 |
| | ✓ | ✓ | 68.8 | 53.6 | 49.3 | **44.6** | 24.9 | **47.3** | **44.0** | 47.5 |
| ✓ | ✓ | ✓ | **70.3** | 51.7 | 49.7 | **44.6** | 27.6 | 46.3 | 43.9 | **47.7** |

Table 6: **Ablation study on the gating residuals in "MoE++ 1B/(16+4)E" model.** All models are trained with a budget of 100B tokens, with the hyper-parameter $\tau$ set to 0.75.

| Methods | Language Tasks | | | | | | | Average |
|---|---|---|---|---|---|---|---|---|
| | PIQA | WinoGrande | ARC-E | HellaSwag (10) | LogiQA | BoolQ (32) | LAMBADA | |
| MoE++ w/o gating residuals | 69.0 | 51.3 | **50.8** | 44.1 | 27.5 | 46.2 | 43.8 | 47.5 |
| MoE++ w/ gating residuals | **70.3** | **51.7** | 49.7 | **44.6** | **27.6** | **46.3** | **43.9** | **47.7** |

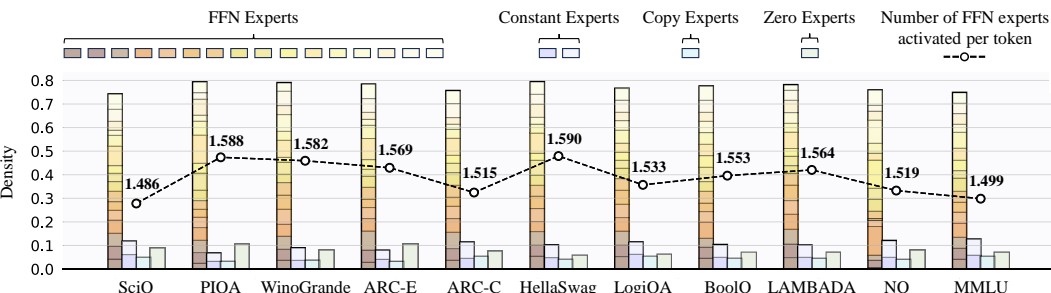

Figure 4: **The visualization of the expert load distribution at the task level.** The result comes from layer 12 of the "MoE++ 7B/(16+4)E" model, with the hyper-parameter $\tau$ set to 0.75.

**Effect of the Number of Constant Experts $n_{const}$.** Compared to zero experts and copy experts, constant experts have trainable vectors, allowing for the addition of multiple constant experts to the MoE++ layer to enhance performance. We gradually increase the number of constant experts $n_{const}$ and provide the ablation results in Fig. 3. We find that average performance initially improves and then decreases. We consider that it is because an increase in constant experts reduces the expert capacity (Eq. 8) of other types of experts. As shown in Fig. 3, given the number of FFN experts $N_{\text{FFN}}$, the number of constant experts $n_{const}$ can be adaptively determined by:

$$n_{const} = \max(\frac{N_{\text{FFN}}}{4} - n_{zero} - n_{copy}, 1),$$ (10)

where $n_{zero}$ represents the number of zero experts. $n_{copy}$ denotes the number of copy experts.

### 4.4 QUALITATIVE ANALYSIS

**Visualization of the Expert Load Distribution at the Task Level.** We provide the visualization of the expert load distribution in Fig. 4. Fig. 4 reveals three key findings: (i) There is a significant variation in the number of FFN experts activated per token across tasks, but it is not necessarily the simpler tasks that activate fewer FFN experts. For example, the ARC Challenge task activates more FFN experts than the ARC Easy task. These results indicate that the MoE++ model assigns experts based on the content of knowledge and complexity at the token level, rather than the overall task difficulty. (ii) Among all expert types, zero experts have the highest average number of activations. Interestingly, simpler tasks show a greater average number of zero expert activations. (iii) We observe that the expert assignments vary significantly across different task topics, indicating that the MoE++ model handles tasks of diverse topics by employing distinct expert assignment patterns. For additional visualizations and a detailed analysis of the expert load distribution, please refer to Appendix D.

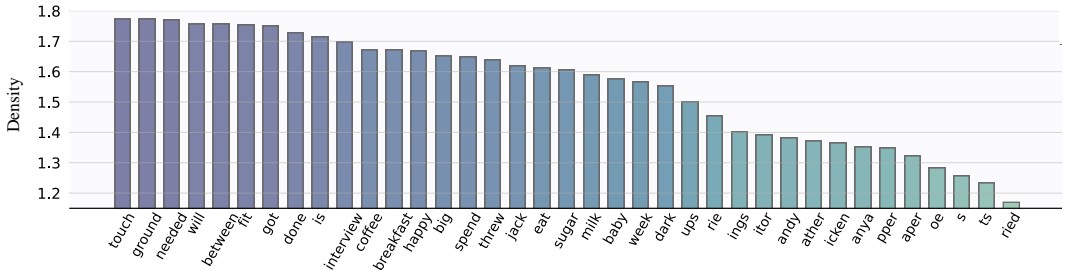

Figure 5: **The visualization of the number of FFN experts activated per token at the token level.** The result comes from the "MoE++ 7B/(16+4)E" model, with the hyper-parameter $\tau$ set to 0.75. We evaluate over 60,000 tokens and average results across all MoE++ layers. Tokenizers often split a word into multiple components, resulting in semantically meaningless tokens such as "icken".

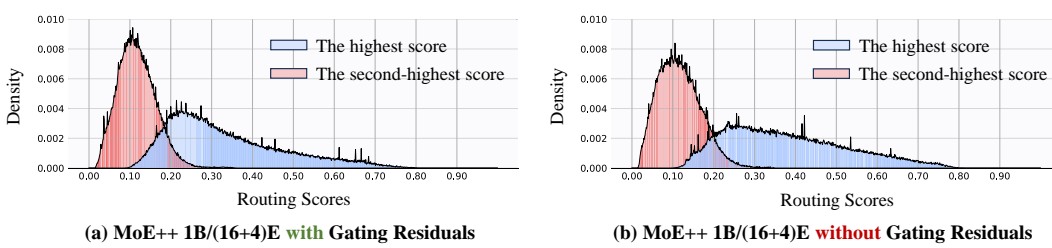

(a) MoE++ 1B/(16+4)E **with** Gating Residuals     (b) MoE++ 1B/(16+4)E **without** Gating Residuals

Figure 6: **The visualization of the impact of gating residuals on routing scores.** We show the highest and second-highest scores of the "MoE++ 1B/(16+4)E" model on the WinoGrande benchmark. All models are trained with a budget of 100B tokens, with the hyper-parameter $\tau$ set to 0.75.

**Visualization of the Number of FFN Experts Activated Per Token at the Token Level.** To explore the average number of FFN expert activations at the token level, we provide the visualization in Fig. 5. The results reveal three observations: (i) Verbs tend to activate a large number of FFN experts. For example, the verb "touch" activates an average of 1.77 FFN experts across all layers, approaching the upper limit of 2. This likely occurs because verbs often convey rich semantic information and frequently interact with nouns to form more complex semantic structures. (ii) Nouns typically activate a moderate number of FFN experts, with most nouns averaging between 1.5 and 1.7 FFN expert activations. (iii) Simple tokens with little semantic tend to activate a small number of FFN experts. For example, word fragments, such as "pper" and "ather", usually activate fewer than 1.5 FFN experts. These findings confirm that MoE++ allows simple tokens to utilize fewer FFN experts, freeing up more FFN experts to focus on challenging tokens.

**Visualization of the Impact of Gating Residuals on Routing Scores.** To better illustrate the impact of the proposed pathway-aware router, we provide a visualization of the effect of gating residuals on routing scores. As shown in Fig. 6, these gating residuals effectively establish connections between different MoE++ layers and reduce the variance of routing scores. Meanwhile, the gating residuals do not change the mean and range of values of the routing scores. Consequently, gating residuals contribute to the stable routing of heterogeneous expert architectures in MoE++.

## 5 CONCLUSION

In this paper, we introduce MoE++, a general and heterogeneous MoE framework that integrates both FFN and zero-computation experts. In contrast to vanilla MoE methods using a fixed mixing mechanism for all tokens, MoE++ optimizes computation allocation by assigning fewer FFN experts to simple tokens, allowing more FFN experts to be dedicated to challenging tokens. Therefore, MoE++ achieves both lower computational overhead and better performance than vanilla MoE. Moreover, since zero-computation experts do not need to be deployed across GPUs, MoE++ is highly deployment-friendly. Extensive experimental results demonstrate that MoE++ not only consistently outperforms vanilla MoE methods but also achieves approximately $1.1 \sim 2.1 \times$ the expert forward throughput of a vanilla MoE model of the same size. Notably, MoE++ is a general framework and can be integrated with any MoE method to enhance both model throughput and performance. We believe MoE++ provides a solid foundation for developing advanced and efficient MoE-related models.

## ACKNOWLEDGEMENTS

This work was supported in part by the Natural Science Foundation of China (No. 62202014, 62332002, 62425101, 62088102), and NUS Start-up Grant A-0010106-00-00. Besides, this work was performed when Peng Jin was an Intern at Skywork AI.

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

# A  APPENDIX

This appendix provides additional discussions (Appendix A), implementation details (Appendix B), details of quantitative evaluations (Appendix C), and more qualitative analysis (Appendix D).

## A  ADDITIONAL DISCUSSIONS

### A.1  EXPERT ARCHITECTURE

Experts in MoE models are typically identical to the standard Feed-Forward Networks (FFNs) used in dense models. Recently, efforts have been made to improve the expert architecture. Deepseek-MoE (Dai et al., 2024) and XMoE (Yang et al., 2024) split the FFN in the dense model into smaller FFNs, reducing the size of each expert while increasing the number of activated experts. PEER (He, 2024) and MH-MoE (Wu et al., 2024) go further by not only reducing the size of experts but also splitting input tokens into smaller units. Although these methods have made some progress, the structure of experts in existing MoE models remains largely based on FFNs, with little exploration of non-FFN or non-parametric experts. To the best of our knowledge, we are the first to propose zero-computation experts for the heterogeneous MoE architecture.

### A.2  THE REAL-WORLD WALL-CLOCK TIME FOR THE ZERO-COMPUTATION EXPERT

In Tab. A, we measure the real-world wall-clock time for the zero-computation expert and FFN expert on a single A100 GPU. As shown in Tab. A, the computational overhead of the zero-computation expert is negligible compared to that of the FFN expert.

Table A: **The real-world wall-clock time for the zero-computation expert and FFN expert.** All tests are conducted on a single A100 GPU.

| Expert Type | # Hidden Size | Number of Tokens | Time (ms) |
|---|---|---|---|
| Zero Expert | 1536 | 2048 | 9.589e-5 |
| Copy Expert | 1536 | 2048 | 1.236e-4 |
| Constant Expert | 1536 | 2048 | 3.206e-2 |
| FFN Expert | 1536 | 2048 | 1.050 |

### A.3  COMPUTATIONAL COSTS OF PATHWAY-AWARE ROUTER

In our proposed pathway-aware router, we add the routing scores from the previous layer to the routing scores predicted by the current layer. Specifically, given the input token $x^j$ of the $j_{th}$ $(j > 1)$ layer with $N$ experts, we use a trainable transformation matrix $W_g^j \in \mathbb{R}^{N \times N}$ to integrate the scores from the previous layer into the current layer:

$$G(\boldsymbol{x}^j) = \begin{cases} \boldsymbol{W}^j \boldsymbol{x}^j, & \text{if } j = 1, \\ \boldsymbol{W}^j \boldsymbol{x}^j + \boldsymbol{W}_g^j G(\boldsymbol{x}^{j-1}), & \text{if } j > 1, \end{cases} \tag{A}$$

where $\boldsymbol{W}^j \in \mathbb{R}^{N \times D}$ is the trainable weight matrix, and $D$ is the hidden size. For example, as shown in Tab. B, in a 16E MoE model with a hidden size of 1536, the additional FLOPs from the gating residuals are only 1% of the FLOPs caused by the original router.

Table B: **Comparison of computational costs between pathway-aware router and original router.**

| Router Type | # Hidden Size | Number of Experts | Number of Tokens | FLOPs | Time (ms) |
|---|---|---|---|---|---|
| Original Router | 1536 | 16 | 2048 | 50,331,648 | 4.264e-2 |
| Pathway-Aware Router | 1536 | 16 | 2048 | 50,888,704 | 4.366e-2 |

### A.4 EFFECTS OF REMOVING CERTAIN COMPONENTS

In our method, the router selects all types of experts, so there is no need to manually remove or modify experts. However, we can also disable some experts. As shown in Tab. C, we find that manually removing some experts would hurt the performance of the model on the WinoGrande benchmark.

Table C: **Effects of removing certain components in "MoE++ 7B/(16+4)E" model.**

| Methods | WinoGrande |
|---|---|
| MoE++ 7B/(16+4)E | 63.1 |
| Remove Zero Expert | 62.0 |
| Remove Copy Expert | 62.2 |
| Remove Constant Expert | 61.9 |
| Remove Zero Expert and Copy Expert | 61.8 |
| Remove Zero Expert and Constant Expert | 61.4 |
| Remove Copy Expert and Constant Expert | 61.7 |
| Remove Zero Expert, Copy Expert, and Constant Expert | 61.0 |

### A.5 LIMITATIONS AND FUTURE WORK

In this section, we delineate the limitations of our work and outline avenues for future research.

**Heterogeneous MoE++ Between Different Layers.** MoE++ implements heterogeneous experts within a single MoE layer. Additionally, as shown in Appendix D, we observe that expert assignment patterns vary more significantly in the shallow and final layers across different tasks, compared to the middle layers. This suggests that the model adapts to tasks primarily through these layers. Future work could explore designing heterogeneous MoE++ configurations across different layers to further enhance the model's adaptability to a wide range of tasks.

**Combining MoE++ with Other Modules.** The current MoE++ method serves as a replacement for the FFN layer in Transformers. Future work could explore integrating other modules, such as combining the attention layer with our MoE++ method.

**The Vulnerabilities of Large Language Models.** The focus of our work is to build advanced and efficient mixture-of-experts Large Language Models (LLMs), and as a consequence, also inherit the vulnerabilities common to LLMs.

- **Hallucination.** Hallucinations in LLMs remain a significant unresolved challenge. These illusory responses can lead to unsupported claims during open-ended conversations, and addressing this issue could greatly accelerate progress in the field. For a deeper analysis of common weaknesses in large LLMs, please refer to Brown et al. (2020); Rae et al. (2021).

- **Long sequence processing.** Transformer-based language models often struggle with generalization when faced with test sequences that are significantly longer than those seen during training. This limitation is especially pronounced in multi-turn conversations, where the model may lose track of the previous context, leading to incorrect responses.

- **Prompt sensitivity.** In-context learning has shown troubling sensitivity to various aspects of demonstrations, such as prompt formats (Zhao et al., 2021). Notably, variations in prompt formats can lead to completely contradictory outputs. Addressing this issue could significantly accelerate progress in the field.

**More Modalities.** Language represents just one facet of communication. Visual and audio information serves to augment and enhance our comprehension of the world (Jin et al., 2024; 2023; 2022; 2025). Future work can explore alternative modalities, such as visual and audio inputs. The incorporation of multiple modalities holds the promise of broadening the spectrum of tasks that the model can address, and it has the potential to enhance their performance by leveraging synergies among these various modalities (Jin et al., 2024).

**More Parameters.** Due to computational constraints, the maximum number of MoE++ model parameters in our experiments is limited to 7B. However, our MoE++ method is highly generalizable and can be scaled to larger models in future research.

## B  IMPLEMENTATION DETAILS

### B.1  DATA DETAILS

Consistent with previous works, we use the tokenizer of LLaMA2, which contains 65,536 vocabulary tokens. It is worth noting that MoE++ is trained exclusively on public datasets, making it accessible for academic research settings. Specifically, we sample from the following datasets according to different sampling probabilities:

- The **RedPajama** (Computer, 2023a) includes training data from seven domains: Common-Crawl, C4, Github, Wikipedia, Books, ArXiv, and StackExchange.

- The **Dolma** (Soldaini et al., 2024), a large and diverse open English text corpus, contains 3 trillion tokens sampled from seven sources, including web pages from Common Crawl, code from The Stack, curated web data from C4 (Raffel et al., 2020), social media conversations from Reddit, academic papers from PeS2o, public domain books from Project Gutenberg, and comprehensive content from Wikipedia and Wikibooks.

- The **Pile** (Gao et al., 2020), an open-source English text corpus for training large language models, includes 22 diverse, publicly available datasets such as Wikipedia, NIH ExPorter, ArXiv, Books3, BookCorpus2, OpenSubtitles, YoutubeSubtitles, and Enron Emails.

Tab. D shows the detailed sample ratios of different open-source datasets. We find that increasing the ratio of high-quality data, such as Books and Wikipedia, during the later stages of training significantly enhances model performance. Consequently, for the "MoE++ 7B/(16+4)E" model in Tab. 4, We increase the ratio of high-quality data for the final 100B tokens. Specifically, this model is trained using strategy 1 for the first 900B tokens and strategy 2 for the last 100B tokens, for a total training budget of 1T tokens. In contrast, for simplicity, all MoE++ and MoE models in Tab. 3 are trained with strategy 1, using a budget of 100B tokens.

Table D: **Sampling ratio of different open-source datasets.** All MoE++ and MoE models in Tab. 3 are trained using strategy 1 with a budget of 100B tokens. In contrast, for the "MoE++ 7B/(16+4)E" model in Tab. 4, strategy 1 is applied for the first 900B tokens, and strategy 2 for the final 100B tokens, resulting in a total training budget of 1T tokens.

|  | Strategy 1 | Strategy 2 |
|---|---|---|
| Redpajama Books | 4.24% | 13.93% |
| Redpajama Wikipedia | 3.50% | 9.03% |
| Redpajama ArXiv | 4.37% | 11.36% |
| Redpajama StackExchange | 3.19% | 9.77% |
| Redpajama C4 | 10.94% | 7.42% |
| Dolma | 61.28% | 41.53% |
| Pile | 12.48% | 6.96% |

### B.2  TRAINING HYPER-PARAMETERS

Tab. E shows the detailed training hyper-parameters. Specifically, the hyper-parameters for MoE++ are selected based on the common practice for dense transformer language models. We replace all FFN layers in the transformer with MoE++ layers and set the Top-K to 2 for every layer, resulting in approximately twice the computation compared to a dense model. The weight $\beta$ for the heterogeneous load balance loss is set to 0.01, and the expert capacity factor $\gamma$ is set to 1.1. MoE++ is trained using the AdamW optimizer (Loshchilov & Hutter, 2017). During training, a weight decay of 0.1 and gradient clipping of 1.0 are applied. All MoE++ (except for the "MoE++ 7B/(16+4)E" with 8-way pipeline parallel) and MoE models in Tab. 3 are trained using strategy 1 with a maximum learning rate of 5e-4 and a batch size of 4 million tokens with a sequence length of 2048. In contrast, for the "MoE++ 7B/(16+4)E" model in Tab. 4, strategy 2 is applied for the first 900B tokens, and strategy 3 for the final 100B tokens, resulting in a total training budget of 1T tokens.

Table E: **Training hyper-parameters.** All MoE++ (except for the "MoE++ 7B/(16+4)E" with 8-way pipeline parallel) and MoE models in Tab. 3 are trained using strategy 1 with a budget of 100B tokens. In contrast, for the "MoE++ 7B/(16+4)E" model in Tab. 4, strategy 2 is applied for the first 900B tokens, and strategy 3 for the final 100B tokens, resulting in a total training budget of 1T tokens.

|  | Strategy 1 | Strategy 2 | Strategy 3 |
|---|---|---|---|
| Training budget | 100B | 900B | 100B |
| Maximum learning rate | 5e-4 | 5e-4 | 1e-4 |
| Final learning rate | 5e-5 | 5e-5 | 1e-5 |
| LR warmup init | 1e-7 | 1e-7 | 1e-7 |
| LR warmup iters | 2000 | 500 | 200 |
| Sequence length | 2048 | 2048 | 2048 |
| Batch size (tokens) | 4M | 4M | 4M |
| Capacity factor $\gamma$ | 1.1 | 1.1 | 1.1 |
| $\beta$ for $\mathcal{L}_b$ | 0.01 | 0.01 | 0.01 |
| Tensor parallel | 1 | 1 | 1 |
| Pipeline parallel | 1 | 8 | 8 |

## C  DETAILS OF QUANTITATIVE EVALUATIONS

We conduct comparative comparisons of MoE++ against vanilla MoE and dense models. The evaluation is performed on multiple key benchmarks using the Eleuther AI Language Model Evaluation Harness[¶] (Gao et al., 2024), a unified framework for testing generative language models across a wide range of tasks. The benchmarks used for evaluation include:

- **ARC** (Clark et al., 2018) is a multiple-choice question-answering resource featuring questions from science exams for grades 3 to 9. It is divided into two partitions: Easy and Challenge, with the latter containing more difficult questions that necessitate reasoning. Most questions offer four answer choices, while less than 1% feature either three or five choices. Additionally, ARC includes a supporting knowledge base with 14.3 million unstructured text passages. We report 0-shot accuracy on ARC Easy (**ARC-E**) and 25-shot accuracy on ARC Challenge (**ARC-C (25)**).

- **LAMBADA** (Paperno et al., 2016) is an open-ended cloze task consisting of approximately 10,000 passages from BooksCorpus, where the objective is to predict a missing target word in the last sentence of each passage. The missing word is always the last word of the final sentence, with no options provided. We report 0-shot accuracy on LAMBADA.

- **LogiQA** (Liu et al., 2020) comprises 8,678 question-and-answer instances that encompass various types of deductive reasoning. The dataset serves as a benchmark for reexamining logical AI within the context of deep learning in NLP. We report 0-shot accuracy on LogiQA.

- **PIQA** (Bisk et al., 2020) is a dataset designed for commonsense reasoning, aimed at evaluating the physical knowledge of current models. We report 0-shot accuracy on PIQA.

- **SciQ** (Welbl et al., 2017) includes 13,679 crowdsourced science exam questions covering subjects such as Physics, Chemistry, and Biology. Each question is presented in a multiple-choice format with four answer options, and for most questions, an additional paragraph provides supporting evidence for the correct answer. We report 0-shot accuracy on SciQ.

- **WinoGrande** (Sakaguchi et al., 2021) is a large-scale dataset comprising 44,000 problems, inspired by the original WSC design but enhanced to increase both its scale and difficulty. We report 0-shot accuracy on WinoGrande.

- **HellaSwag** (Zellers et al., 2019) is a challenging dataset designed to evaluate commonsense Natural Language Inference (NLI), which proves difficult for state-of-the-art models but poses no significant challenge for humans. We report the accuracy for the 10-shot HellaSwag (**HellaSwag (10)**).

- **MMLU** (Hendrycks et al., 2021) is a benchmark designed to assess models' knowledge acquired during pretraining, making it more challenging and human-like in evaluation.

---

[¶]https://github.com/EleutherAI/lm-evaluation-harness

It covers 57 subjects across STEM, humanities, social sciences, and more, ranging from elementary to advanced professional levels. The benchmark tests both world knowledge and problem-solving skills, with subjects spanning traditional areas like math and history to specialized fields such as law and ethics, offering a comprehensive tool for identifying model blind spots. We report the accuracy for the 5-shot MMLU (**MMLU (5)**).

- **Natural Questions (NQ)** (Kwiatkowski et al., 2019) is a question-answering dataset based on real, anonymized Google queries. Annotators label long and short answers (or null if no answer is found) from Wikipedia pages in the top 5 search results. The dataset includes 307,373 training examples, 7,830 development examples, and 7,842 test examples with 5-way annotations. We report the exact match score for 32-shot Natural Questions (**NQ (32)**) to measure the factual knowledge in the model.

- **BoolQ** (Clark et al., 2019) is a question-answering dataset consisting of 15,942 yes/no questions. These questions are naturally occurring, and generated in unprompted and unconstrained contexts. Each example is provided as a triplet of (question, passage, and answer), with the page title optionally included as additional context. We report the accuracy for the 32-shot BoolQ (**BoolQ (32)**).

## D  ADDITIONAL QUALITATIVE ANALYSIS

### D.1  THE NUMBER OF ZERO-COMPUTATION EXPERTS ACTIVATED PER TOKEN

To explore the average number of zero-computation expert activations at the token level, we provide the visualization in Fig. A. The results reveal three observations: (i) Zero experts specialize in processing the simplest tokens, such as single letters, spaces, and punctuation. (ii) Copy experts focus on handling word fragments, such as "hol" and "ts", while also managing single letters and punctuation. (iii) Constant experts specialize in processing word fragments and simple words, such as "it" and "She". These results demonstrate that MoE++ automatically assigns tokens to experts based on their complexity and the capabilities of experts. Specifically, zero experts output only empty features, copy experts replicate the input as output, and constant experts introduce additional trainable vectors to adjust the output. Therefore, MoE++ assigns the simplest tokens to zero experts, moderately simple tokens to copy experts, and slightly more complex tokens to constant experts.

### D.2  THE EXPERT LOAD DISTRIBUTION ACROSS ALL LAYERS

To explore the expert load distribution across all layers in the MoE++ model across different tasks, we provide the visualizations of the expert load distribution at the task level in Fig. B, Fig. C, Fig. D, Fig. E, and Fig. F. These visualizations reveal several interesting findings:

- We observe a correlation in expert load across different layers, particularly between adjacent layers. For example, when layer $j$ activates a large proportion of FFN experts, there is a high likelihood that layer $j + 1$ will also activate FFN experts in a similarly large proportion.

- We find that expert assignment patterns in the shallow and final layers vary more significantly across tasks compared to the middle layers. This suggests that the model primarily adapts to different tasks through its shallow and final layers, rather than the middle layers. Future work could focus on designing more complex structures in these layers to enhance the model's adaptability to diverse tasks.

- There is a significant variation in the number of FFN experts activated per token across tasks, but it is not necessarily the simpler tasks that activate fewer FFN experts. For example, the ARC Challenge task usually activates more FFN experts than ARC Easy. These results indicate that the MoE++ model assigns experts based on the content of knowledge and complexity at the token level, rather than the overall task difficulty.

- Among all expert types, zero experts have the highest average number of activations, with simpler tasks showing a greater average number of activations. For example, the ARC Easy task activates more zero experts than the ARC Challenge task. This indicates that the level of zero expert activation may serve as an indicator of task difficulty for the model.

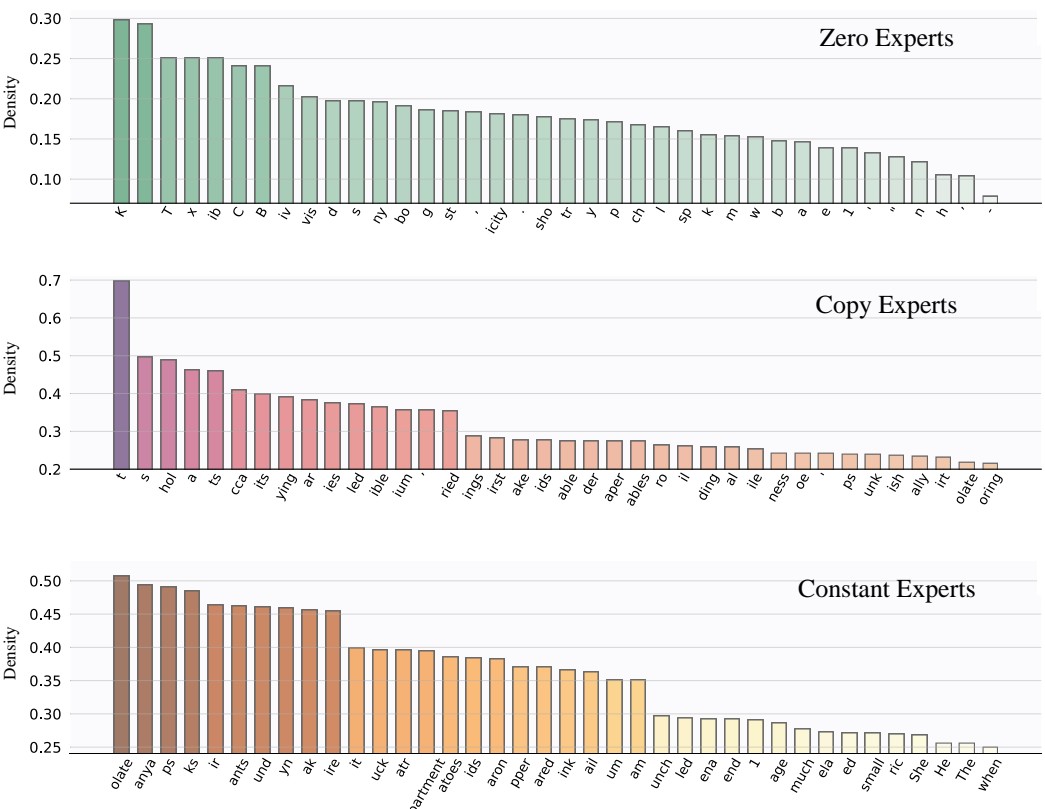

Figure A: **The visualization of the number of zero-computation experts (zero experts, copy experts, and constant experts) activated per token at the token level.** The result comes from the "MoE++ 7B/(16+4)E" model, with the hyper-parameter $\tau$ set to 0.75. We evaluate over 60,000 tokens and average results across all MoE++ layers. Tokenizers often split a word into multiple components, resulting in semantically meaningless tokens such as "hol".

- We also observe that the expert assignments vary significantly across different task topics for all layers in the MoE++ model, indicating that the MoE++ model handles tasks of diverse topics by employing distinct expert assignment patterns.

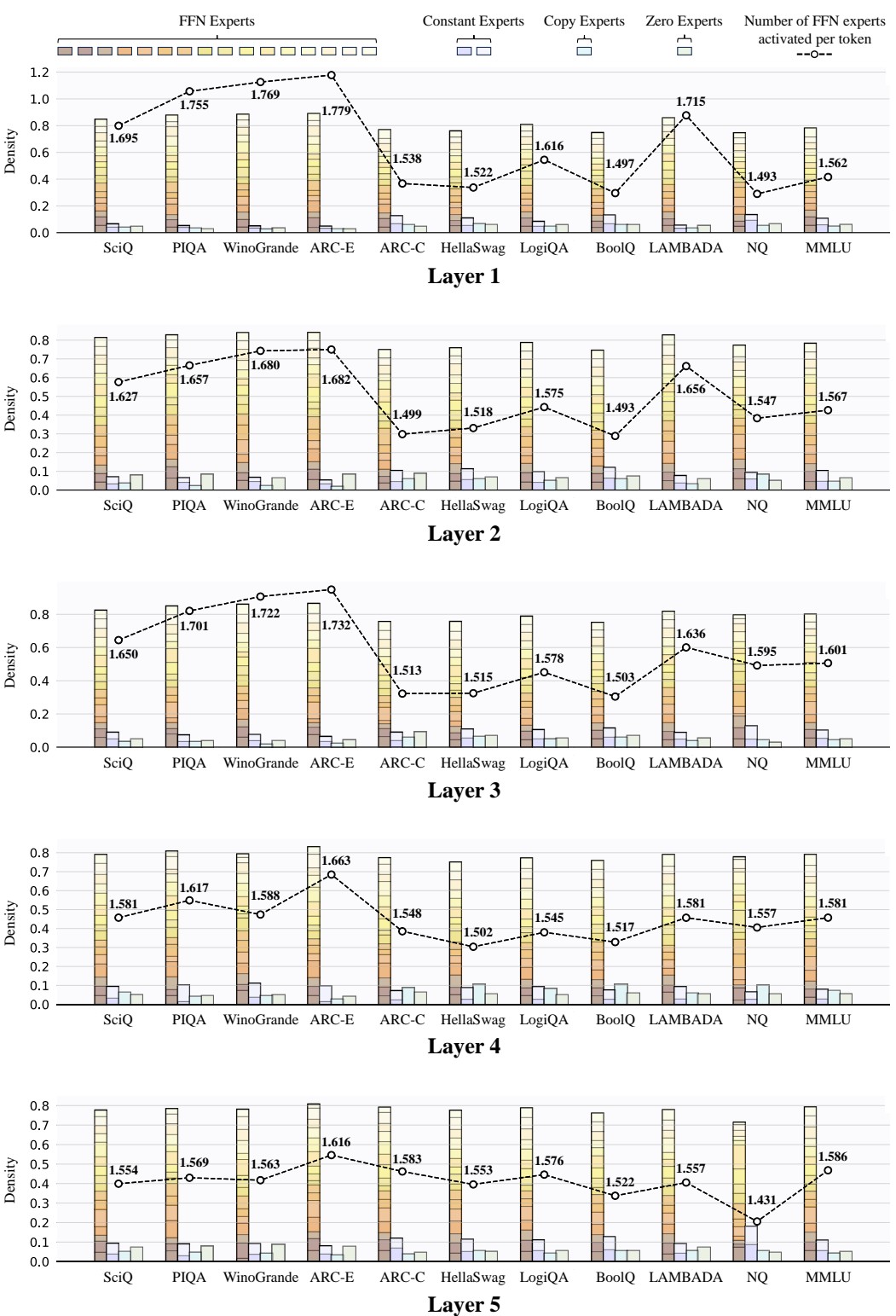

Figure B: **The visualization of the expert load distribution at the task level.** The results come from layer 1 to layer 5 of the "MoE++ 7B/(16+4)E" model, with the hyper-parameter $\tau$ set to 0.75.

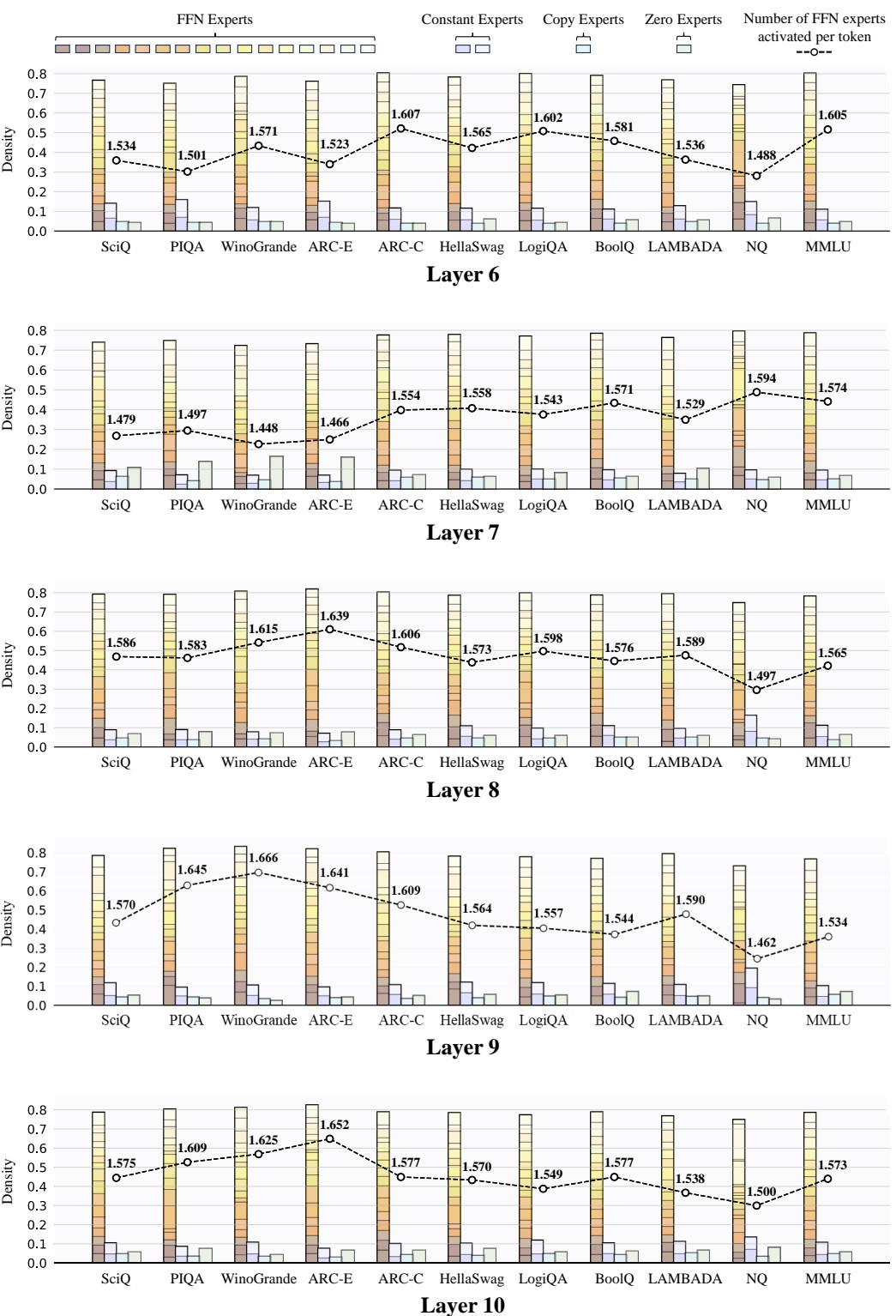

Figure C: **The visualization of the expert load distribution at the task level.** The results come from layer 6 to layer 10 of the "MoE++ 7B/(16+4)E" model, with the hyper-parameter $\tau$ set to 0.75.

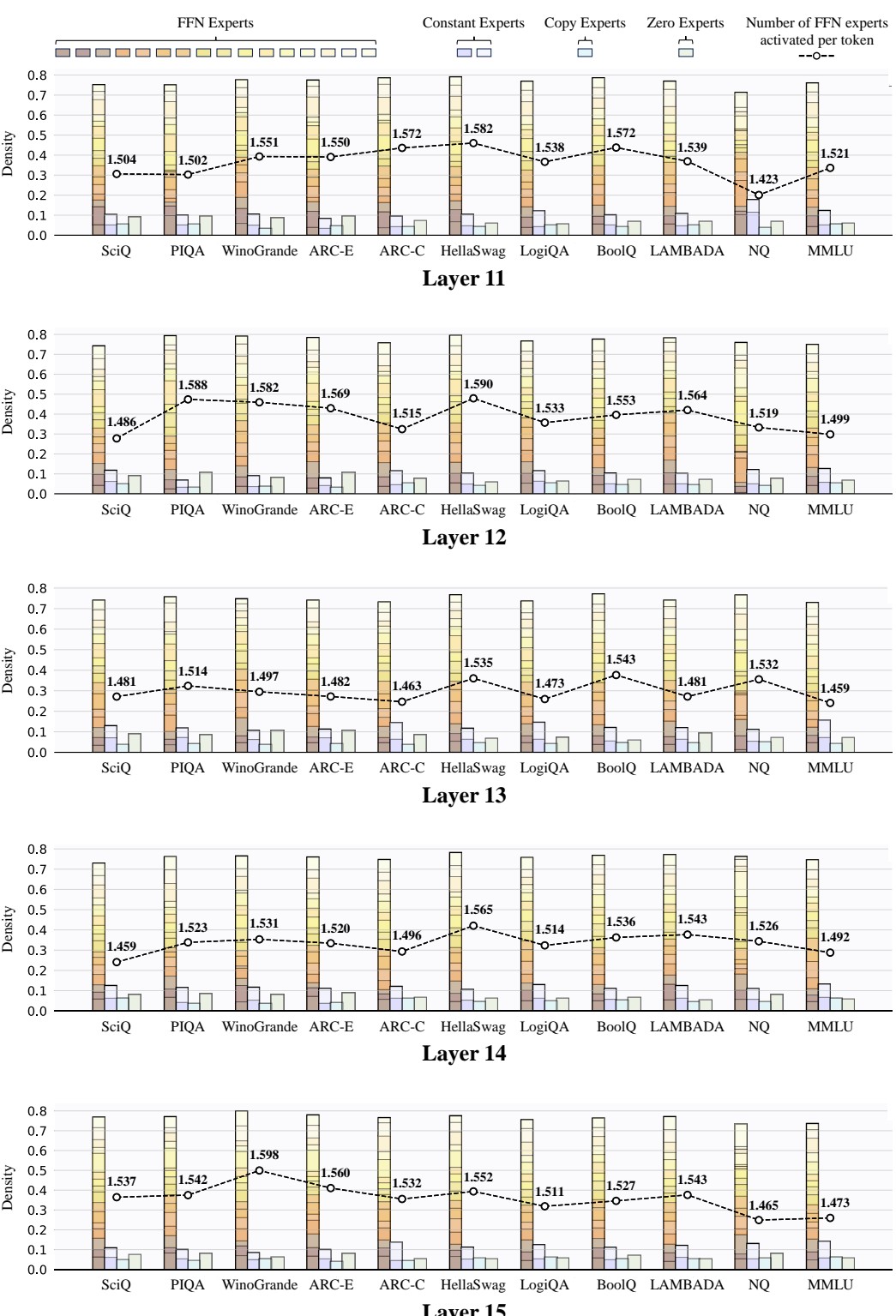

Figure D: **The visualization of the expert load distribution at the task level.** The results come from layer 11 to layer 15 of the "MoE++ 7B/(16+4)E" model, with the hyper-parameter $\tau$ set to 0.75.

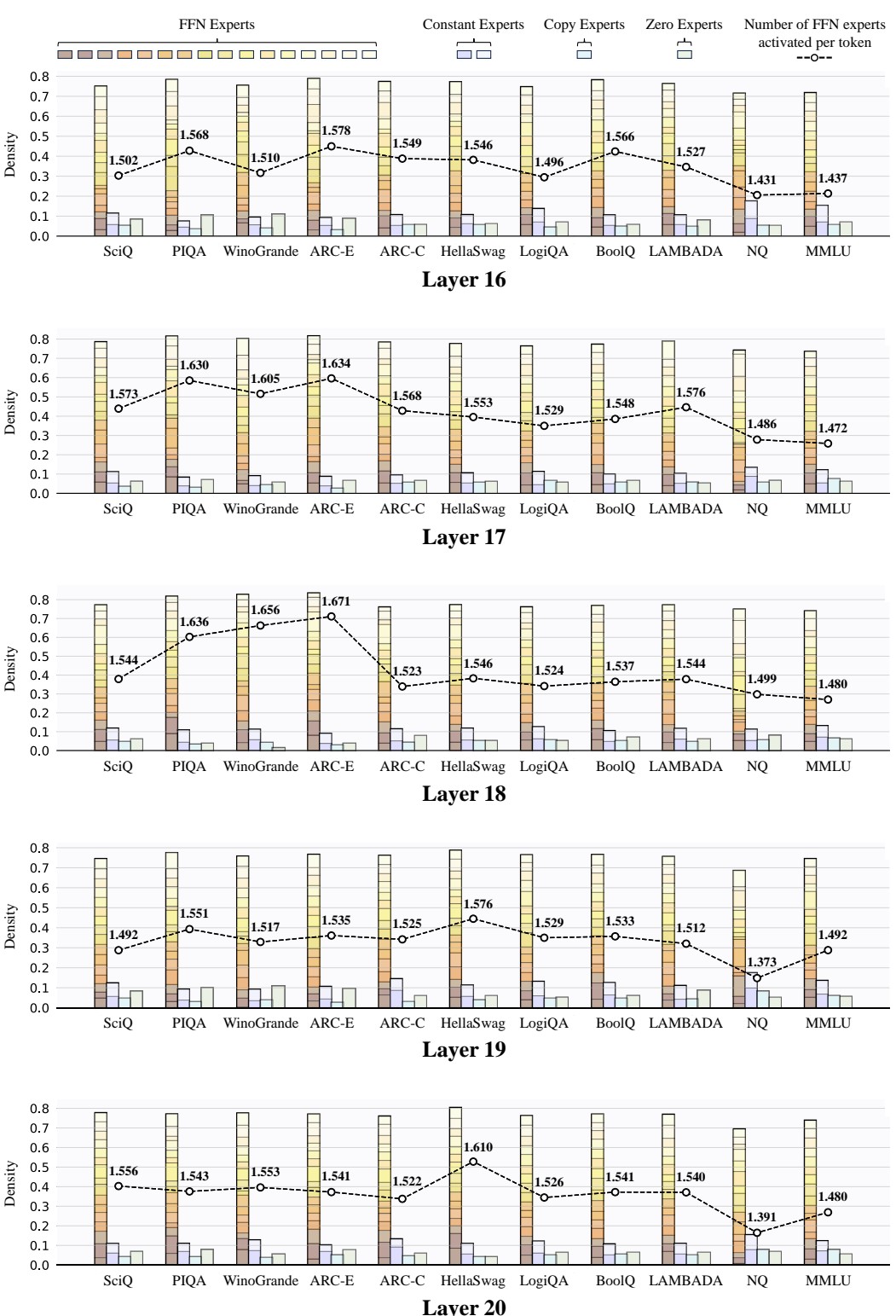

Figure E: **The visualization of the expert load distribution at the task level.** The results come from layer 16 to layer 20 of the "MoE++ 7B/(16+4)E" model, with the hyper-parameter $\tau$ set to 0.75.

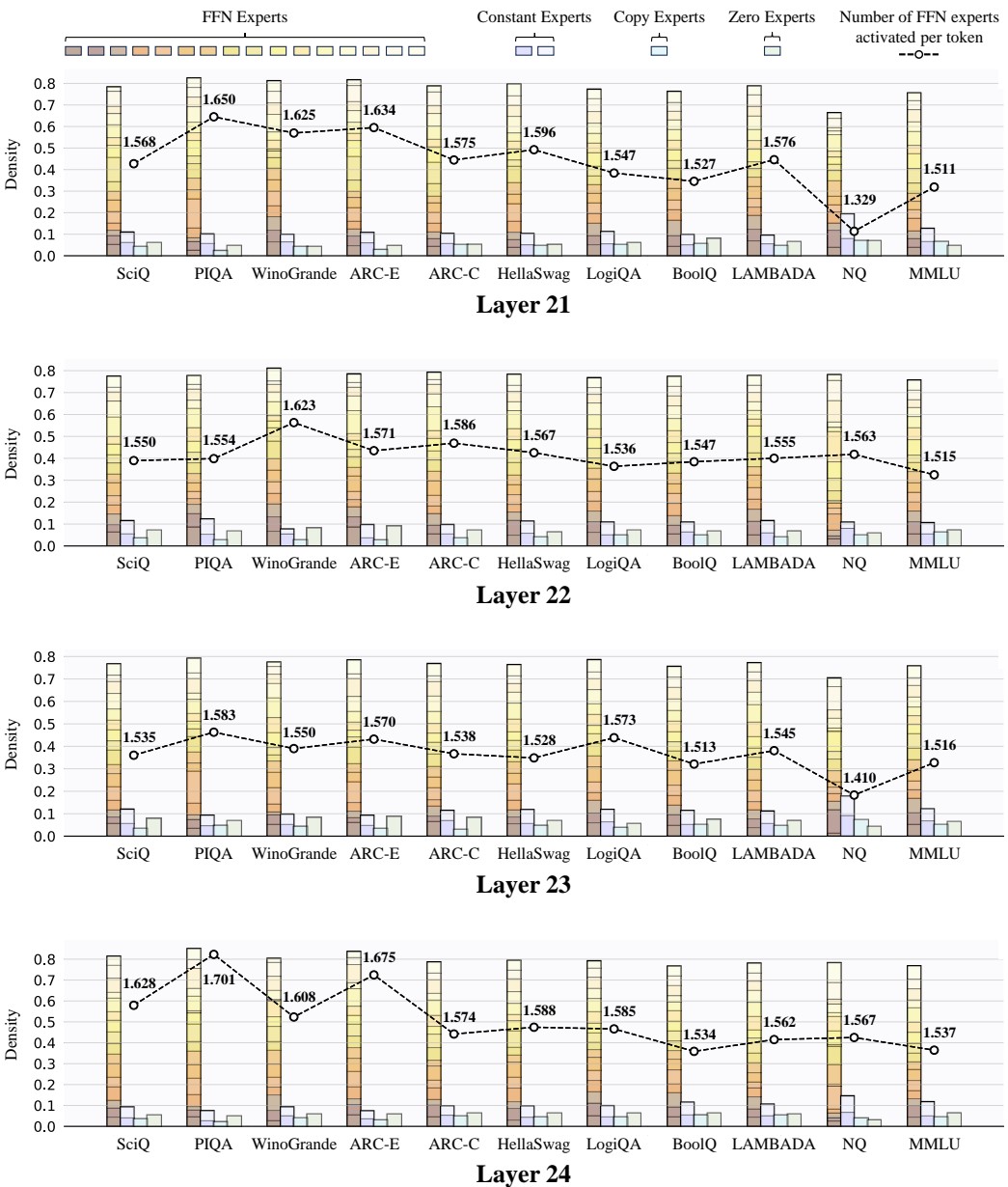

Figure F: **The visualization of the expert load distribution at the task level.** The results come from layer 21 to layer 24 of the "MoE++ 7B/(16+4)E" model, with the hyper-parameter $\tau$ set to 0.75.

