# OpenReview forum: "MoE++: Accelerating Mixture-of-Experts Methods with Zero-Computation Experts"
_ICLR.cc/2025/Conference — ICLR 2025 Oral_

### Official Review · Reviewer_LU5B · 2024-10-31

**Soundness:** 3
**Presentation:** 4
**Contribution:** 3
**Rating:** 8
**Confidence:** 5

**Summary:**

This paper introduces MoE++, a general and heterogeneous MoE framework that uses a fixed mixing mechanism for all tokens and optimizes computation allocation by assigning fewer FFN experts to simple tokens, allowing more FFN experts to be dedicated to challenging tokens. Extensive experimental results demonstrate the lower computational overhead and better performance of MoE++ than vanilla MoE.

**Strengths:**

- This work improves the existing MoE framework in terms of both efficiency (throughput) and effectiveness (performance), making it impactful for real-world applications.
- This paper is well-written and insightful, with clear motivation and illustrations.
- Figures and tables are clear and easy to read.

**Weaknesses:**

- While Table 1 shows the “complexity between the proposed MoE++ and MoE” and zero-computation experts enjoy a complexity of 0, they still likely lead to some extra computation overhead. Therefore, this work lacks real-world wall-clock time demonstrations of these zero-computation expert operators, especially regarding a batch of tokens.
- This work introduces three types of zero-computation experts (i.e. zero, copy, and constant) but provides minimal justification for this specific set. From Table 5, we can see they only conducted basic combinatorial ablations.

**Questions:**

- How are these zero-computation experts specialized to input tokens? Specifically, it would be great to show some examples of tokens allocated to zero, copy, and constant experts, respectively.

---

> ### Author Response · Authors · 2024-11-20
> **Responses to the Reviewer LU5B [1/2]**
>
> We sincerely appreciate your thoughtful comments and your recognition that "this work improves the existing MoE framework in terms of both efficiency (throughput) and effectiveness (performance), making it impactful for real-world applications," as well as that "this paper is well-written and insightful, with clear motivation and illustrations."
> Below, we provide detailed responses to your questions.
>
> **Q1: This work lacks real-world wall-clock time demonstrations of these zero-computation expert operators, especially regarding a batch of tokens.**
>
> **A1:** Thanks for your insightful advice. Following your suggestion, we measured the real-world wall-clock time for these zero-computation expert operations on a single A100 GPU.
>
> As shown in the table below, the computational overhead of the zero-computation expert is negligible compared to that of the FFN expert. We have added these results in the revised manuscript (Table A on Page 15).
>
> | Expert Type     | # Hidden Size | **Number of Tokens** | **Time (ms)** |
> |:----------------|:-------------:|:--------------------:|:-------------:|
> | Zero Expert     |     1536      |         2048         |   9.589e-5    |
> | Copy Expert     |     1536      |         2048         |   1.236e-4    |
> | Constant Expert |     1536      |         2048         |   3.206e-2    |
> | FFN Expert      |     1536      |         2048         |     1.050     |
>
>
> **Q2: This work introduces three types of zero-computation experts (i.e. zero, copy, and constant) but provides minimal justification for this specific set. From Table 5, we can see they only conducted basic combinatorial ablations.**
>
> **A2:** Thanks for your valuable comments. In our method, we consider the redesigned expert architecture should satisfy specific criteria:
>
> * It should be as streamlined as possible to process simple tokens efficiently;
> * To ensure a fair comparison with the vanilla MoE, the new expert should introduce an almost negligible number of parameters.
>
> Guided by these principles, we propose three types of zero-computation experts: the zero expert, copy expert, and constant expert, which correspond to discard, skip, and replace operations, respectively.
>
> Nevertheless, our MoE++ method can be easily extended to include more expert types, such as FFNs with varying hidden feature dimensions, although this poses a challenge for fair comparisons with standard MoE methods. We will also continue to explore the possibility of more expert types in future work.
>
> In Table 5, we explore as many different combinations of experts as possible. Specifically, as shown in the table below, we examine all possible expert combinations.
>
> | Methods                                             | **Average Performance** |
> |:----------------------------------------------------|:-----------------------:|
> | Baseline                                            |          46.6           |
> | with Zero Expert                                    |          47.0           |
> | with Copy Expert                                    |          47.2           |
> | with Constant Expert                                |          47.4           |
> | with Zero Expert + Copy Expert                      |          47.2           |
> | with Zero Expert + Constant Expert                  |          47.3           |
> | with Copy Expert + Constant Expert                  |          47.5           |
> | with Zero Expert + Copy Expert + Constant Expert    |        **47.7**         |
>
>
> Due to the time constraints for rebuttal, we are unable to provide results for FFNs with different dimensions. However, we will continue to improve our method as per your suggestion.

---

> ### Author Response · Authors · 2024-11-20
> **Responses to the Reviewer LU5B [2/2]**
>
> **Q3: How are these zero-computation experts specialized to input tokens? Specifically, it would be great to show some examples of tokens allocated to zero, copy, and constant experts, respectively.**
>
> **A3:** Thanks for your valuable advice. Following your suggestion, we have added the visualization of the number of zero-computation experts (zero experts, copy experts, and constant experts) activated per token at the token level (Figure A on Page 19).
>
> These visualizations reveal several interesting findings:
>
> * Zero experts specialize in processing the simplest tokens, such as single letters, spaces, and punctuation.
> * Copy experts focus on handling word fragments, such as "hol" and "ts", while also managing single letters and punctuation.
> * Constant experts specialize in processing word fragments and simple words, such as "it" and "She".
>
> These results demonstrate that our proposed MoE++ automatically assigns tokens to experts based on their complexity and the capabilities of experts. Specifically, zero experts output only empty features, copy experts replicate the input as output, and constant experts introduce additional trainable vectors to adjust the output. Therefore, MoE++ assigns the simplest tokens to zero experts, moderately simple tokens to copy experts, and slightly more complex tokens to constant experts.
>
> Besides, the table below provides examples of tokens handled by each type of expert.
>
> |  Expert Type     |                         Tokens                          |
> |:----------------|:-------------------------------------------------------:|
> | Zero Expert     |        "K", " ", "T", "x", "ib", "C", "B", "iv"         |
> | Copy Expert     |    "t", "s", "hol", "a", "ts", "cca", "its", "ying"     |
> | Constant Expert | "olate", "anya", "ps", "ks", "ir", "ants", "und", "yn"  |

---

> ### Author Response · Authors · 2024-11-23
>
> Dear Reviewer
>
> Could we kindly inquire if the responses have satisfactorily tackled your concerns, or if there is a need for further experiment and visualization? Your commitment to reviewing our work is immensely appreciated. We sincerely thank you for your prompt and insightful review of our paper. Your comment is immensely appreciated and undoubtedly helps improve the quality of our work.

---

> > ### Comment · Reviewer_LU5B · 2024-11-24
> >
> > Thanks for the clear answers. My concerns were adequately addressed, so I will keep my rating at 8.

---

> > > ### Author Response · Authors · 2024-11-24
> > > **Sincere appreciation**
> > >
> > > We sincerely thank you for your prompt and insightful review of our paper. Your comment is immensely appreciated and undoubtedly helps improve our work. Thanks again for taking the time and effort when handling our paper.

---

### Official Review · Reviewer_FDic · 2024-11-04

**Soundness:** 3
**Presentation:** 3
**Contribution:** 3
**Rating:** 8
**Confidence:** 3

**Summary:**

This paper proposes the addition of a new class of experts in the MoE architecture called "zero-computation" experts. The zero-computation type experts include a zero expert, a copy expert, and constant experts. Each of these experts is extremely computationally lightweight relative to a standard FFN expert. The authors demonstrate that incorporating this expert type is both GPU friendly due to the low overhead and can improve performance and throughput.

**Strengths:**

The authors provide a fairly simple and effective addition to the standard MoE architecture which effectively allows more heterogeneous computing in different layers with minimal overhead. The paper is written clearly and a thorough set of ablations are performed.

**Weaknesses:**

The empirical gains don't seem to be too significant and can only start to be seen at large scale ~7B parameters.

**Questions:**

Are the baseline MoE models also trained with residual gating? For each model, many experts are activated in each layer? I could not really tell.

---

> ### Author Response · Authors · 2024-11-20
> **Responses to the Reviewer FDic**
>
> We sincerely appreciate your thoughtful comments, especially noting that we "provide a fairly simple and effective addition to the standard MoE architecture which effectively allows more heterogeneous computing in different layers with minimal overhead" and that "the paper is written clearly and a thorough set of ablations are performed."
> Below, we provide detailed responses to your questions.
>
> **Q1: The empirical gains don't seem to be too significant and can only start to be seen at large scale ~7B parameters.**
>
> **A1:** Thanks for your insightful comments. We explain it to you from the following three aspects:
>
> * Our method achieves an average improvement of 1.9 on small-scale models, such as the 1B model, across nine benchmarks. Given that our method significantly enhances the model's training and inference speed without increasing the number of parameters, we consider this average accuracy gain of 1.9 to be meaningful.
> * In addition to improving the model performance, our method is also valuable to improve the efficiency of the model. We show that our method achieves approximately 1.1$\sim$2.1$\times$ the expert forward throughput of a vanilla MoE model of the same size.
> * In practical applications, MoE models are typically large in scale. Therefore, it is acceptable that our method achieves greater improvements on models with larger-scale parameters.
>
>
> **Q2: Are the baseline MoE models also trained with residual gating?**
>
> **A2:** Thanks for your detailed review of our manuscript. The baseline MoE models are trained without residual gating.
>
> The table below shows the results of the baseline MoE model using residual gating. As shown in the table below, we find that the improvement of residual gating on the vanilla MoE model is not significant.
>
> We consider this may be because all experts in the vanilla MoE are structured the same. In contrast, our proposed MoE++ contains heterogeneous experts, making the design of the router more critical compared to vanilla MoE. As a result, the benefits of residual gating are more pronounced in MoE++.
>
>
> | Methods                               | **Average Performance** |
> |:--------------------------------------|:-----------------------:|
> | Vanilla MoE without residual gating   |          46.6           |
> | Vanilla MoE with residual gating      |        **46.7**         |
>
>
> **Q3: For each model, how many experts are activated in each layer?**
>
> **A3:** Thanks for your detailed review of our manuscript. In all models, two experts are activated at each layer.
>
> We have clarified the model settings in the revised manuscript. Specifically, we have added the following note to the caption of Table 2 (which presents the model settings): "In all models, two experts are activated at each layer."

---

> ### Author Response · Authors · 2024-11-23
>
> Dear Reviewer
>
> Would it be possible for us to kindly ascertain if the provided responses have satisfactorily tackled any concerns you may have had and if further explanations or clarifications are needed? Your generous investment of time and effort in the evaluation of our work is truly commendable. We extend our heartfelt gratitude for your insightful commentary and the considerable time you have devoted to reviewing our paper.

---

> > ### Comment · Reviewer_FDic · 2024-11-24
> >
> > Thank you for the detailed responses. Based on the discussion I have raised my score to an 8.

---

> > > ### Author Response · Authors · 2024-11-25
> > > **Sincere appreciation**
> > >
> > > Thank you for your helpful feedback. Your expertise and careful review have greatly improved our work. We truly appreciate the time and effort you took to give such a detailed review of our paper.

---

### Official Review · Reviewer_SD7n · 2024-11-04

**Soundness:** 2
**Presentation:** 3
**Contribution:** 2
**Rating:** 8
**Confidence:** 4

**Summary:**

This paper introduces the MoE++, which adds zero-computation experts to enhance the efficiency of computation.
It utilizes zero-computation experts to minimize overhead by allocating fewer resources to basic tokens and concentrating on complex ones. MoE++ enhances expert selection stability by employing pathway-aware routing with gating residuals, resulting in higher performance and throughput than conventional MoE models at a reduced computational cost.

**Strengths:**

With the proper hyper-parameters, MoE++ introduces zero-computation experts who can reduce the computational load by bypassing or simplifying processing for certain tokens, leading to efficient resource use.
These heterogeneous experts with suitable routing designs can improve the model's efficiency and effectiveness.

**Weaknesses:**

1. Parameters such as $\tau$, which regulate token allocation between zero-computation experts and original, may complicate model tuning, as the performance and burden distribution are sensitive to them.
2. The pathway-aware routing with gating residuals adds complexity to the expert selection process, which may require careful tuning for optimal results.
3. The dynamic routing mechanism can still result in load imbalances, especially under diverse data distributions, which may lead to underutilized or overloaded experts that affect efficiency.

**Questions:**

1. How do individual components, like zero experts or gating residuals, contribute to MoE++'s overall performance?
2. Could certain components be removed or modified for specific use cases?

---

> ### Author Response · Authors · 2024-11-20
> **Responses to the Reviewer SD7n [1/3]**
>
> We sincerely appreciate your thoughtful comments and your recognition that our proposed zero-computation experts "with suitable routing designs can improve the model's efficiency and effectiveness."
> Below, we provide detailed responses to your questions.
>
> **Q1: Parameters such as $\tau$, which regulate token allocation between zero-computation experts and original, may complicate model tuning, as the performance and burden distribution are sensitive to them.**
>
> **A1:** Thanks for your valuable comments. In our method, $\tau$ can be used to balance computing costs and performance.
> Specifically, a smaller $\tau$ means that more tokens are assigned to the zero-computation experts with negligible computing costs, resulting in higher throughput. Conversely, a larger $\tau$ means fewer tokens are allocated to the zero-computation experts and generally have better performance.
>
> As shown in the tables below, we find that $\tau=0.75$ strikes a good balance between computing costs and model performance.
> Therefore, we recommend setting the hyper-parameter $\tau$ to 0.75 by default.
>
> |                   | # Activated Params | # $\tau$ | **Expert Throughput Increase** | **Average Performance** |
> |:------------------|:------------------:|:--------:|:------------------------------:|:-----------------------:|
> | MoE 0.6B/8E       |     0.2B/0.6B      |    -     |               -                |          44.3           |
> | MoE++ 0.6B/(8+4)E |  $\leq$0.2B/0.6B   |   0.10   |             164.5%             |          43.4           |
> | MoE++ 0.6B/(8+4)E |  $\leq$0.2B/0.6B   |   0.25   |             92.7%              |          44.9           |
> | MoE++ 0.6B/(8+4)E |  $\leq$0.2B/0.6B   |   0.50   |             38.2%              |          44.9           |
> | MoE++ 0.6B/(8+4)E |  $\leq$0.2B/0.6B   |   0.75   |             25.2%              |        **45.6**         |
> | MoE++ 0.6B/(8+4)E |  $\leq$0.2B/0.6B   |   1.00   |             19.1%              |          44.5           |
>
> |                  | # Activated Params | # $\tau$ | **Expert Throughput Increase** | **Average Performance** |
> |:-----------------|:------------------:|:--------:|:------------------------------:|:-----------------------:|
> | MoE 1B/16E       |      0.2B/1B       |    -     |               -                |          45.3           |
> | MoE++ 1B/(16+4)E |   $\leq$0.2B/1B    |   0.10   |             111.2%             |          45.8           |
> | MoE++ 1B/(16+4)E |   $\leq$0.2B/1B    |   0.25   |             58.7%              |          45.9           |
> | MoE++ 1B/(16+4)E |   $\leq$0.2B/1B    |   0.50   |             30.1%              |          46.0           |
> | MoE++ 1B/(16+4)E |   $\leq$0.2B/1B    |   0.75   |             22.1%              |          46.1           |
> | MoE++ 1B/(16+4)E |   $\leq$0.2B/1B    |   1.00   |             15.2%              |        **47.2**         |
>
> |                  | # Activated Params | # $\tau$ | **Expert Throughput Increase** | **Average Performance** |
> |:-----------------|:------------------:|:--------:|:------------------------------:|:-----------------------:|
> | MoE 2B/32E       |     0.2B/2B        |    -     |               -                |          45.9           |
> | MoE++ 2B/(32+8)E |   $\leq$0.2B/2B    |   0.10   |             63.5%              |          46.3           |
> | MoE++ 2B/(32+8)E |   $\leq$0.2B/2B    |   0.25   |             44.3%              |          47.0           |
> | MoE++ 2B/(32+8)E |   $\leq$0.2B/2B    |   0.50   |             28.3%              |          46.4           |
> | MoE++ 2B/(32+8)E |   $\leq$0.2B/2B    |   0.75   |             21.8%              |          47.3           |
> | MoE++ 2B/(32+8)E |   $\leq$0.2B/2B    |   1.00   |             15.7%              |        **48.6**         |
>
> |                  | # Activated Params | # $\tau$ | **Expert Throughput Increase** | **Average Performance** |
> |:-----------------|:------------------:|:--------:|:------------------------------:|:-----------------------:|
> | MoE 7B/16E       |        1.2B/7B     |    -     |               -                |          52.3           |
> | MoE++ 7B/(16+4)E |   $\leq$1.2B/7B    |   0.75   |             27.8%              |        **53.1**         |

---

> ### Author Response · Authors · 2024-11-20
> **Responses to the Reviewer SD7n [2/3]**
>
> **Q2: The pathway-aware routing with gating residuals adds complexity to the expert selection process, which may require careful tuning for optimal results.**
>
> **A2:** Thanks for your thoughtful comments. The computing costs of gating residuals are negligible.
>
> In our proposed pathway-aware router, we add the routing scores from the previous layer to the routing scores predicted by the current layer. Specifically, given the input token ${x}^{j}$ of the $j_{th}$ ($j$>1) layer with $N$ experts, we use a trainable transformation matrix ${W}^j_{g} \in \mathbb{R}^{N\times N}$ to integrate the scores from the previous layer into the current layer: $G({x}^{j})={W}^{j}{x}^{j}+{W}^{j}_{g}G({x}^{j-1})$.
>
>
> For example, in a 16E MoE model with a hidden size of 1536, the additional FLOPs from gating residuals are only **1\%** of the FLOPs caused by the original router.
>
> |                      | # Hidden Size | # The Number of Experts | Number of Tokens | **FLOPs** | **Time (ms)** |
> |:---------------------|:-------------:|:-----------------------:|:----------------:|:---------:|:-------------:|
> | Original Router      |     1536      |           16            |       2048       |  50,331,648   |   4.264e-2    |
> | Pathway-Aware Router |     1536      |           16            |       2048       |  50,888,704   |   4.366e-2    |
>
>
> **Q3: The dynamic routing mechanism can still result in load imbalances, especially under diverse data distributions, which may lead to underutilized or overloaded experts that affect efficiency.**
>
> **A3:** Thanks for your valuable comments. We address your concern from the following two perspectives:
>
> * **First, load imbalance is a common issue for all MoE methods.** Specifically, model parameters are dynamically activated in MoE. MoE methods balance the load across experts when handling a lot of data. However, load imbalance can still happen during inference on a single sample. This means some experts might always be activated, while others are not used.
> * **Second, our proposed MoE++ faces a lower risk of load imbalance compared to the vanilla MoE methods.** Specifically, since zero-computation experts have negligible parameters, we can deploy all zero-computation experts on each GPU. This setup eliminates the significant communication overhead and load imbalance typically caused by FFN experts being distributed across multiple GPUs.
>
>
> **Q4: How do individual components, like zero experts or gating residuals, contribute to MoE++'s overall performance?**
>
> **A4:** Thanks for your helpful comments. We show in Table 5 how the components, like zero experts, contribute to MoE++'s overall performance. Table 6 shows how gating residuals contribute to MoE++'s overall performance.
>
> As shown in the table below, we find that constant experts improve the model more than zero experts and copy experts. We consider that it is due to the increased flexibility that constant experts provide in handling tokens. Specifically, zero experts output only empty features, copy experts replicate the input as output, and constant experts introduce additional trainable vectors to adjust the output. Our full model, which incorporates all three types of zero-computation experts, achieves the best performance, demonstrating their benefit for language models.
>
> | Methods                                             | **Average Performance** |
> |:----------------------------------------------------|:-----------------------:|
> | Baseline                                            |          46.6           |
> | with Zero Expert                                    |          47.0           |
> | with Copy Expert                                    |          47.2           |
> | with Constant Expert                                |          47.4           |
> | with Zero Expert + Copy Expert                      |          47.2           |
> | with Zero Expert + Constant Expert                  |          47.3           |
> | with Copy Expert + Constant Expert                  |          47.5           |
> | with Zero Expert + Copy Expert + Constant Expert    |        **47.7**         |
>
> As shown in the table below, we find that gating residuals effectively establish connections between different MoE++ layers, thereby ensuring stable routing.
>
> | Methods                     | **Average Performance** |
> |:----------------------------|:-----------------------:|
> | MoE++ without gating residuals    |          47.5           |
> | MoE++ with gating residuals       |        **47.7**         |

---

> ### Author Response · Authors · 2024-11-20
> **Responses to the Reviewer SD7n [3/3]**
>
> **Q5: Could certain components be removed or modified for specific use cases?**
>
> **A5:** Thanks for your insightful comments. In our method, all types of experts are selected by the router, so there is no need to manually remove or modify experts.
>
> Following your suggestion, we try to disable some experts on the WinoGrande benchmark. As shown in the table below, we find that manually removing some experts would hurt the performance of the model.
>
> | Methods                                              | **Average Performance** |
> |:-----------------------------------------------------|:-----------------------:|
> | MoE++                                                |          63.1           |
> | Remove Zero Expert                                   |       62.0 (-1.1)       |
> | Remove Copy Expert                                   |       62.2 (-0.9)       |
> | Remove Constant Expert                               |       61.9 (-1.2)       |
> | Remove Zero Expert and Copy Expert                   |       61.8 (-1.3)       |
> | Remove Zero Expert and Constant Expert               |       61.4 (-1.7)       |
> | Remove Copy Expert and Constant Expert               |       61.7 (-1.4)       |
> | Remove Zero Expert, Copy Expert, and Constant Expert |       61.0 (-2.1)       |

---

> ### Author Response · Authors · 2024-11-23
>
> Dear Reviewer
>
> May we kindly inquire if the provided responses have adequately addressed any questions you might have had? If there remains a requirement for further explanations or clarifications? We wish to express our sincere gratitude for your meticulous evaluation and for generously investing a significant amount of your time in reviewing our paper. Your feedback would be greatly valued.

---

> > ### Comment · Reviewer_SD7n · 2024-11-26
> >
> > Thank you for the detailed responses. Based on your response, I have increased my score to an 8.
> > Additionally, I do think these experiments should be added to the paper's Appendix.

---

> > > ### Author Response · Authors · 2024-11-26
> > > **Sincere appreciation**
> > >
> > > Thank you very much for your thoughtful review of our paper. Your comments are greatly appreciated and truly help improve the quality of our work. Following your suggestion, we have added these experiments to the Appendix:
> > >
> > > * Comparison of computational costs between pathway-aware router and original router (Table B on Page 15).
> > > * Effects of removing certain components (Table C on Page 16).
> > >
> > > We truly appreciate your time and effort in reviewing our paper.

---

### Author Response · Authors · 2024-11-20
**Global Response**

We sincerely thank all PCs, SACs, ACs, and Reviewers for their time and efforts when handling our paper.

All reviewers appreciate the contributions of our method:
* All reviewers pointed out that our work improves the existing MoE framework in terms of **both efficiency (throughput) and effectiveness (performance)**.
* Reviewers FDic and LU5B all highlighted that our paper is **written clearly** and **extensive experiments are performed**.
* Reviewer LU5B commented that our method is **impactful for real-world applications**.

As suggested by the reviewers, we have revised the manuscript as follows:
* The real-world wall-clock time for zero-computation experts and FFN experts (Table A on Page 15).
* Comparison of computational costs between pathway-aware router and original router (Table B on Page 15).
* Effects of removing certain components (Table C on Page 16).
* The visualization of the number of zero-computation experts (zero experts, copy experts, and constant experts) activated per token at the token level (Figure A on Page 20).
* Modifying the caption of Table 2 on Page 6.

We highlight all the modifications using red color.

---

### Meta-Review · Area_Chair_fEwo · 2024-12-18

**Metareview:**

This paper proposes MoE++, a heterogeneous MoE framework that enhances both the efficiency and effectiveness of MoE methods by integrating FFN and zero-computation experts (zero, copy, and constant experts). This design reduces computing overhead by dynamically assigning simpler tokens to zero-computation experts, improves performance by focusing FFNs on challenging tokens, and eliminates GPU communication overhead, making it deployment-friendly. Experimental results show that MoE++ outperforms vanilla MoE models of the same size while achieving up to 2.1× faster expert forward throughput.

Most reviewers acknowledge the efficacy and simplicity of the proposed algorithms. Most concerns lie in real running time and ablation analysis. During the rebuttal, the authors did a great job of addressing these concerns. We recommend an acceptance.

**Additional Comments On Reviewer Discussion:**

During the rebuttal, the authors did a great job of addressing these concerns. The authors are highly encouraged to include the discussions accordingly.

---

### Decision · Program_Chairs · 2025-01-22

Accept (Oral)